# Advances in Prognostic Methylation Biomarkers for Prostate Cancer

**DOI:** 10.3390/cancers12102993

**Published:** 2020-10-15

**Authors:** Dilys Lam, Susan Clark, Clare Stirzaker, Ruth Pidsley

**Affiliations:** 1Epigenetics Research Laboratory, Genomics and Epigenetics Division, Garvan Institute of Medical Research, Sydney, New South Wales 2010, Australia; d.lam@garvan.org.au (D.L.); s.clark@garvan.org.au (S.C.); c.stirzaker@garvan.org.au (C.S.); 2St. Vincent’s Clinical School, University of New South Wales, Sydney, New South Wales 2010, Australia

**Keywords:** DNA methylation, epigenetics, biomarkers, circulating DNA, cfDNA, prostate cancer, early detection, prognosis

## Abstract

**Simple Summary:**

Prostate cancer is a major cause of cancer-related death in men worldwide. There is an urgent clinical need for improved prognostic biomarkers to better predict the likely outcome and course of the disease and thus inform the clinical management of these patients. Currently, clinically recognised prognostic markers lack sensitivity and specificity in distinguishing aggressive from indolent disease, particularly in patients with localised, intermediate grade prostate cancer. Thus, there is major interest in identifying new molecular biomarkers to complement existing standard clinicopathological markers. DNA methylation is a frequent alteration in the cancer genome and offers potential as a reliable and robust biomarker. In this review, we provide a comprehensive overview of the current state of DNA methylation biomarker studies in prostate cancer prognosis. We highlight advances in this field that have enabled the discovery of novel prognostic genes and discuss the potential of methylation biomarkers for noninvasive liquid-biopsy testing.

**Abstract:**

There is a major clinical need for accurate biomarkers for prostate cancer prognosis, to better inform treatment strategies and disease monitoring. Current clinically recognised prognostic factors, including prostate-specific antigen (PSA) levels, lack sensitivity and specificity in distinguishing aggressive from indolent disease, particularly in patients with localised intermediate grade prostate cancer. There has therefore been a major focus on identifying molecular biomarkers that can add prognostic value to existing markers, including investigation of DNA methylation, which has a known role in tumorigenesis. In this review, we will provide a comprehensive overview of the current state of DNA methylation biomarker studies in prostate cancer prognosis, and highlight the advances that have been made in this field. We cover the numerous studies into well-established candidate genes, and explore the technological transition that has enabled hypothesis-free genome-wide studies and the subsequent discovery of novel prognostic genes.

## 1. Introduction

Prostate cancer (PCa) is the most commonly diagnosed noncutaneous cancer in men and one of the leading causes of cancer death in males. Globally, 1,276,106 new cases were diagnosed in 2018 alone [1,2], and this number is projected to rise by approximately 80%, to more than two million new cases a year, by 2040 [3,4,5]. Currently, PCa diagnosis is achieved through assessment of blood prostate-specific antigen (PSA) levels, digital rectal examination (clinical T-stage) and histological examination of needle biopsies (Gleason Score (GS)/ISUP Grade) [6]. PSA-based screening was introduced in the late 1980s, and has significantly increased the early detection of localized disease [7,8,9]. Diagnosis at this early organ-confined stage of disease is crucial as it is potentially curable by radical prostatectomy (RP), a procedure to surgically remove the whole prostate gland. While this is curative for most prostate cancers, approximately 30% of patients treated by RP experience biochemical recurrence (BCR) [10], and 17–22% of these relapsed patients progress to metastatic-lethal PCa [11,12,13]. There is therefore a need to identify the men at high risk of metastatic progression, so that additional interventions can be offered earlier (e.g., adjuvant therapies such as chemotherapy and radiotherapy) [14].

On the other hand, many men diagnosed with PCa have an indolent form of the disease, which is characterised by slow progression with no eventual clinical manifestation [15,16]. For these men, RP represents an overtreatment given the risk of unnecessary side effects and compromised quality of life [17,18]. Thus, strategies such as ‘watchful waiting’ and ‘active surveillance’ have emerged for men diagnosed with low grade disease, in which regular monitoring is used to detect tumour progression, with the aim of delaying RP until it is clinically necessary [19]. However, 13–45% of low-risk men on active surveillance exhibit a PSA rise and progress to surgery or other treatments [20,21,22], indicating that they may have been inappropriately assigned to monitoring, and should have been treated earlier.

PCa has a heterogeneous clinical course which makes it challenging to decide the most appropriate treatment or monitoring strategy for individual patients [23]. There is a major unmet clinical need for specific prognostic biomarkers that can accurately differentiate indolent from aggressive tumours that are likely to metastasise and lead to lethal disease [24]. The ability to identify the risk of progression at initial diagnosis would inform decisions about personalized treatment and/or monitoring strategies, as well as the use of adjuvant therapy, for improved clinical management and enhanced outcomes for PCa patients (Figure 1).

### 1.1. Current Clinicopathological Prognostic Markers

The existing scoring systems and nomograms that are used to identify patients at high risk of progression are based solely on routine clinical and pathological variables at the time of diagnosis or surgery, including preoperative (pre-op) PSA levels, GS/ISUP grade scoring, tumour staging (clinical or pathological T-stage) and margin status (Figure 1) [11,25,26,27,28,29,30]. These tools, whilst useful in the diagnostic assessment of the tumour, lack sensitivity and specificity in classifying the risk of an individual patient’s disease [6,31,32,33]. One problem with the current prognostic approach is that a number of scoring systems exist, with no single accepted standard system being used in clinical practice [29,34]. Furthermore, these prognostic variables are based on histological assessment, which may not capture the underlying biology of a tumour and its potential to progress to an aggressive, lethal cancer. Advances in molecular profiling techniques mean that we can now access another layer of biological information in tumours, including early molecular changes that may precede histologically visible alterations [23,35]. Much research is now being dedicated to investigating whether molecular information can better inform clinical decisions about individualised treatment and/or monitoring strategies, as well as providing new biological insights to guide the development of new therapies [35].

### 1.2. Molecular Biomarkers for Prognosis

There is increasing evidence for the value of prognostic molecular biomarkers to complement existing standard clinicopathological markers. Molecular prognostication of PCa has been investigated in various contexts, including at the level of genetic, epigenetic and gene expression alterations [36,37], from both tumour tissue [37,38] and liquid biopsies [39,40] (Figure 1). To date, the major success stories of prognostic molecular biomarkers are the commercialised tissue-based tests centred on panels of gene expression signatures. These tests include Prolaris (Myriad Genetics, Salt Lake City, UT, USA), Oncotype DX Prostate (Genomic Health, Redwood City, CA, USA) and Decipher (GenomeDX Biosciences, Vancouver, BC, Canada) [37]. The Prolaris test determines risk of progression (BCR, cancer-specific death) by measuring a proliferation signature of 31 cell cycle progression transcripts [41], whilst Oncotype DX predicts adverse pathology (high grade/stage disease) or poor outcome (BCR) based on 12 genes [42]. Decipher is based on 22 markers which have been trained to predict early metastasis and aggressive PCa [43]. Whilst these tests demonstrate the potential for molecular biomarkers, they have yet to be integrated into the standard clinical routine following the initial diagnosis [37].

### 1.3. DNA Methylation Biomarkers

DNA methylation is one of the earliest, most stable and frequent alterations in the cancer genome and has been extensively investigated as a source of molecular biomarkers [36,44,45,46]. DNA methylation is an epigenetic modification, in which a methyl group is added to the cytosine base of a cytosine-guanine (CpG) dinucleotide. It is associated with gene regulation and function, with promoter-associated clusters of marks, termed CpG islands, often linked to gene silencing [47,48]. In the context of PCa, aberrant DNA methylation is a key feature observed during early tumorigenesis, as well as in its progression and metastatic development [49,50,51], and occurs at a much higher frequency and more consistently than genetic mutations [52]. Additionally, DNA methylation has been shown to outperform gene expression in detecting cancer from prostate biopsies [53]. The feasibility of using DNA methylation as a biomarker is further supported by the fact that DNA is more stable than RNA [54] and its methylation patterns are retained following long-term storage of clinical material, including as formalin-fixed paraffin-embedded tissue (FFPET). DNA methylation assays can also be easily integrated into routine clinical practice as many diagnostic labs already have the infrastructure in place, due to their similarity with DNA-sequence-based biomarker approaches [55]. There is currently one validated epigenetic test commercially available, ConfirmMDx (MDxHealth, Irvine, CA, USA), designed for diagnostic rather than prognostic purposes, that uses the methylation profile of three genes (*APC*, *RASSF1*, *GSTP1*) to detect cancer in histologically negative biopsies [56]. PCa-derived aberrant DNA methylation patterns have also been detected in liquid biopsies such as blood and urine samples, paving the way for the development of noninvasive molecular tests [57,58,59,60].

## 2. Current State of Prognostic Methylated Biomarkers

A plethora of studies have been conducted over the last two decades investigating DNA methylation-based biomarkers to aid PCa prognosis. The majority of these studies have interrogated primary prostate tumours extracted from RP tissue, whilst others have used prostate tissue from core or needle biopsies, transurethral resection of the prostate (TURP) specimens, as well as tumour-adjacent and benign nonneoplastic prostate specimens. The studies reviewed in this section have been primarily performed on RP tissue, and studies that have used other types of prostate specimens will be noted accordingly. Initial studies in this field, limited by the laboratory techniques available, took the traditional a priori approach of examining genes that had been implicated in biological pathways of PCa. These studies used targeted methylation profiling techniques including methylation-specific PCR (MSP) [61], quantitative methylation-specific PCR (qMSP) [62], pyrosequencing [63,64] and mass spectrometry (MassARRAY EpiTYPER, Agena Bioscience, San Diego, California, USA) [65] to assess the methylation profile of a specific gene of interest (Figure 2). Figure 2 lists other targeted approaches that have also been used to assess methylation in cancer, including ddPCR [66], COBRA [67], high resolution melt curve [68] and headloop MSP [69,70,71]. With recent advances in the technological capabilities to interrogate the methylome more broadly, the field has transitioned into conducting hypothesis-free genome-wide screens for novel prognostic methylation biomarkers. The candidate and genome-wide studies reviewed below use a range of outcome measures to assess the prognostic value of methylated genes, including low vs. high grade cancers, localised vs. advanced disease, and clinical outcomes such as BCR, metastatic relapse and PCa-specific death (PCa death). Importantly, the clinical outcome most often studied is BCR, defined by an increase in serum PSA levels post-RP. However, there is increasing evidence that BCR is not a sufficient indicator of progression to aggressive lethal disease, with metastatic relapse instead being the clinically relevant endpoint for predicting survival [72,73]. This requires long-term follow-up (≥15 years) for metastatic relapse and PCa death to manifest [16], which many study populations lack, thus reducing their ability to fully evaluate and assess molecular biomarkers of PCa prognosis.

### 2.1. Candidate (A Priori) Markers

To date, DNA methylation of over 60 candidate genes has been investigated. To summarise the top candidate markers with the greatest prognostic evidence, this review focuses on genes that have been investigated in at least three studies and used well-defined prognostic outcomes. The studies reporting on these genes are detailed in Table 1. The majority of these studies were performed on RP-derived tissue, when other types of tissue were used (for example, needle biopsies of prostate tissue or urine) this is specified below or within Table 1. Figure 3 visualises the 20 genes examined in these studies and specifies which of these were found to have potential prognostic value in univariate (U) and/or multivariate (M) statistical models (adjusted for clinicopathological variables) of prognosis. Below we discuss the genes that have been most extensively studied and validated across independent studies: *GSTP1*, *APC*, *RARB*, *PITX2*, *CCND2* and *PTGS2*.

#### 2.1.1. GSTP1

The *glutathione S-transferase pi* gene (*GSTP1*) is the most well-studied DNA methylation biomarker of PCa, particularly in diagnosis [100]. It encodes glutathione *S-*transferase, a detoxifying enzyme and tumour suppressor involved in drug metabolism and protecting DNA from oxidative damage [101]. Hypermethylation (increased methylation) of *GSTP1* is observed frequently in PCa tissue but rarely in histologically negative prostate tissues [102]. We identified 15 candidate prognostic biomarker studies that have studied *GSTP1* methylation, with its prognostic value validated in 8 of these studies (Table 1). The earliest of these studies assessed *GSTP1* methylation using qMSP in a cohort of GS 7 (3 + 4) patients (*n* = 74), and reported that *GSTP1* hypermethylation was significantly associated with time to progression (any of BCR, metastatic relapse and/or PCa death) in univariate analysis, and as an independent predictor in multivariate analysis with other candidate genes [75]. Subsequent studies of RP tissue, using qMSP, have again found *GSTP1* to be an independent prognostic factor. Briefly, Maldonado et al. used a large cohort (*n* = 452) to show that *GSTP1* methylation was a significant independent prognostic factor of progression in a multivariate model adjusting for age at surgery, pre-op PSA, surgery year, surgical margins, pathological T-stage and GS; but only in samples from early, organ-confined disease (*n* = 183) [92]. Litovkin et. al. found trichotomised *GSTP1* methylation to be an independent prognostic predictor (when adjusted for GS, pathological T-stage and pre-op PSA levels) of clinical failure (see Table 1 for definition) in two cohorts (Training: *n* = 147, Validation: *n* = 71) of high-risk PCa patients [94]. In a study of non-neoplastic tissue adjacent to the prostate tumour from PCa patients with follow-up up to 24 years (*n* = 157), the presence of *GSTP1* methylation increased risk of PCa death by almost 3-fold, and remained an independent prognostic factor in multivariate models in combination with *APC* methylation, GS, age at diagnosis, year of diagnosis, source of tumour tissue and methylation in matched tumour tissue [88]. Another study assessing PCa death, using pyrosequencing of TURP tissue (*n* = 367), also found univariate associations with *GSTP1* methylation [90].

Two small studies with BCR as the clinical endpoint observed associations with *GSTP1* methylation: (1) in a univariate analysis, using sextant biopsy cores (*n* = 83) [77] and (2) as part of a multigene signature, using RP tissue (*n* = 41) [78]. Another small study (*n* = 64) investigating the broader outcome of recurrence (any of BCR, local recurrence or metastatic relapse) observed associations with *GSTP1* methylation at 3 CpG units in univariate analysis only [81]. Importantly, of the 15 studies assessing *GSTP1* methylation as a prognostic biomarker, 7 studies did not find any prognostic value in *GSTP1* methylation in the prediction of BCR [74,76,79,89,93], low vs. high GS cancers [84] or PCa death [77,82].

#### 2.1.2. APC

*APC* is a tumour suppressor gene which encodes the adenomatous polyposis coli (APC) protein, with a known role in the cellular processes of tumourigenesis [103]. Hypermethylation of *APC* is observed in PCa tumours [104] and a number of studies have demonstrated its prognostic potential (Table 1). In two of these studies, *APC* hypermethylation was included in multivariate models alongside *GSTP1* methylation to predict progression in GS 7 patients [75] and PCa death (using DNA from non-neoplastic adjacent tissue) [88]. *APC* methylation has also been observed to be an individual methylation marker of BCR [77,96] and PCa death [77,82]. In the study by Henrique et al., methylation levels of 5 genes (*APC*, *CCND2*, *GSTP1*, *RARB*, *RASSF1*) were quantified by qMSP of DNA extracted from sextant biopsies (*n* = 83), of which *APC* was the only gene significantly associated with both BCR and PCa death in univariate and multivariate analyses with other clinicopathological factors. [77]. A separate study that used MSP to quantify *APC*, *RUNX3*, *GSTP1* reported that *APC* was the only independent prognostic gene in the prediction of PCa death across two cohorts of RP patients, one before PSA-testing was widespread (1980s cohort: *n* = 216) and one after the introduction of PSA-testing (1990s cohort: *n* = 243); adjusting for source of tumour tissue, GS and follow-up duration [82]. One study looking at BCR examined *APC* methylation using nested MSP in benign prostate specimens (needle biopsy or TURP) from patients who eventually developed PCa (*n* = 353) and found associations between *APC* and risk of BCR in White patients only (*n* = 206), adjusting for age, tumour stage, GS, PSA level and treatment type [96]. Two other studies of *APC* methylation levels reported only univariate associations with risk of BCR [78] and PCa death in TURP tissues [90]. It should be noted that, while these studies provide evidence for the potential prognostic value of *APC* methylation, a number of other studies did not observe such associations with predicting risk of BCR [74,79,89], low vs. high GS [84], clinical failure [94], progression [92] and PCa death within 10 years of diagnosis [95].

#### 2.1.3. RARB

The *RARB* gene encodes the retinoic acid receptor beta protein, a nuclear transcriptional regulator important in cellular signalling in cell growth and differentiation processes, and often silenced and hypermethylated in PCa [105]. In total, 5 of the 12 studies examining *RARB* methylation have reported its potential prognostic utility, of which 2 studies have observed *RARB* methylation levels to be an independent prognostic variable in multivariate models including other clinicopathological factors [94,96]. Briefly, the qMSP study by Litovkin et al., examining prediction of clinical failure in high-risk patients, found dichotomised *RARB* methylation, significant in univariate and multivariate analyses (adjusted for GS, pathological T-stage and PSA) in a small validation cohort (*n* = 41), but not in the training cohort (*n* = 71) [94]. In their more recent study, methylation of the *RARB* promoter region significantly increased risk of BCR in African American patients (*n* = 147), but only when another gene was methylated (*APC*, *CTNND2*, *RASSF1* or *MGMT*) and no inflammation was present in the prostate specimen [96]. Other studies reported only univariate associations between methylation and prognosis with the clinical endpoints of PCa death [90] and BCR if hypermethylated alongside four or more candidate genes [78] or in RP patients with GS ≤ 7 [89]. Other studies assessing BCR [76,93], recurrence or progression [75,81,92] and PCa death [77] found no associations with *RARB* methylation. 

#### 2.1.4. PITX2

*PITX2* encodes the paired-like homeodomain transcription factor 2, induced by the WNT pathway to activate growth regulating genes required for cell-type specific proliferation [106]. Aberrant *PITX2* methylation has been observed in multiple tumour types including breast [107] and PCa [80]. Of nine studies on *PITX2* methylation in Table 1, all but one study, which assessed methylation in benign prostate specimens [96], reported significant associations with risk of progression. The first study by Weiss and colleagues, observed *PITX2* methylation, quantified by qMSP, as the strongest and only independent predictor of BCR, providing additional prognostic information to existing clinicopathological factors of GS and pathological T-stage [80]. A subsequent study of *n* = 476 patients confirmed the association between *PITX2* hypermethylation, also quantified by qMSP, and increased BCR risk in a multivariate model with GS, pathological T-stage, surgical margin, age and PSA levels [83]. Another study, using samples from the same cohort as [83], plus a smaller training cohort (training cohort: *n* = 157, testing cohort: *n* = 523) observed that *PITX2* methylation added prognostic information to GS, pathological T-stage and surgical margins in the prediction of BCR in multivariate cox analysis. Vanaja et al. constructed a methylation score consisting of 11 CpG units across 5 genes (from the EpiTYPER MassARRAY platform, see Table 1) including sites in the *PITX2* promoter region to predict recurrence within 5 years, in a model combined with GS, pre-op PSA, seminal vesicle involvement and margin status, achieving an AUC of 0.852 (Sensitivity/Specificity = 80/81.2%) [81]. In a more recent study investigating PCa death as the clinical endpoint in a large cohort of patient-derived TURP tissue (*n* = 385), a prognostic model was built on 6 methylation biomarkers (see Table 1) including *PITX2*, and was able to improve on the sensitivity of the Cancer of the Prostate Risk Assessment (CAPRA) score to predict aggressive PCa at 10 years follow-up with an AUC of 0.74 (Table 1) [98]. A smaller study of risk of PCa death in patients with GS ≤ 7 (*n* = 135, median follow-up of 15 years, TURP tissue specimens), also using pyrosequencing, reported a significant association with increased *PITX2* methylation levels [91]. Two additional studies from the same lab likewise reported univariate associations between *PITX2* methylation and BCR, with no multivariate analysis with clinicopathological factors conducted [97,99]. Additional analyses showed that combinations of both *PITX2* and *PITX3* methylation were associated with BCR [97] and that there was a strong correlation between *PITX2* methylation and ISUP grade group in core needle biopsy specimens [99].

#### 2.1.5. CCND2 and PTGS2

*CCND2* and *PTGS2* have been studied in eight and six studies respectively, with four studies reporting modest evidence of potential prognostic value for each gene. Hypermethylation of *CCND2* has been reported as an independent prognostic marker of clinical failure [94], and for prediction of progression [75] and PCa death [90,98] in combination with other markers. Three small studies (each *n* ≤ 60) observed associations between higher *PTGS2* methylation and increased risk of BCR [74,76,78] with one study reporting a nine-fold increased risk when combined with *CD44* methylation [76], whilst a larger two-cohort study reported associations with clinical failure in the training cohort only [94].

#### 2.1.6. Other Candidate Genes

Other candidate genes that have been investigated in at least three studies include *RASSF1*, *TIG1*, *EDNRB*, *MGMT*, *MDR1*, *CDKN2A*, *TIMP3*, *CDH1*, *PDLIM4*, *DPYS*, *MAL*, *SLIT2*, *SFN* and *HSPB1* (Table 1). Methylation of four of these genes (*MGMT*, *CDKN2A*, *TIMP3*, *CDH1*) was not found to have any prognostic utility, whilst *EDNRB*, *MDR1* and *PDLIM4* methylation was only reported to have significant univariate associations with disease risk in one study each (Figure 3). Surprisingly, *RASSF1*, frequently hypermethylated in PCa and part of the ConfirmMDX panel [56], has only been reported to have associations with risk of BCR in two small studies [77,93]. *TIG1* was not observed to be an independent prognostic factor alone, only in combination with other genes for prediction of PCa death in low-to-intermediate-risk patients [98]. Pairwise combinations of high *SLIT2*, *SFN* and *SERPINB5* methylation were able to classify high from low GS patients in random forest modelling of a small cohort (*n* = 48) [84]. *DPYS*, *MAL* and *HSPB1* emerge as potential prognostic biomarkers, with *HSPB1* in particular validating as an independent prognostic factor of PCa death in the three studies it has been investigated in [86,90,98]. *SERPINB5* and *AIM1*, the only markers found to have significant associations with progression [95] and overall survival [85] have only been investigated in two small independent studies each thus far, and require further validation in larger cohorts. And finally, new candidate genes from different biological pathways have been explored recently (e.g., *PD1−*, *PD-L1*, *CDO1*, *TFF3*, *ZNF660*) [108,109,110,111,112] and also require further validation.

In summary, the evidence for the prognostic value of the most extensively studied candidate genes (e.g., *GSTP1*, *APC*, *RARB*, *CCND2*, *PTGS2*) is conflicting, potentially due to differences in study designs including diversity in sample type, cohort size, clinical endpoints examined, methylation profiling methodologies, analytical approach and clinicopathological factors adjusted for in multivariate analyses. Thus far, *PITX2* methylation has shown the most robust evidence of providing additive prognostic information to traditional clinicopathological markers, in particular in the prediction of BCR progression.

### 2.2. Genome-Wide Prognostic Biomarker Discovery Studies

Technological advancements in microarray and next-generation sequencing technologies over the last decade have enabled hypothesis-free, genome-wide screening for new prognostic methylation biomarkers. Platforms that have been used for epigenome-wide screens include restriction enzyme-based, capture-based and microarray-based platforms, together with next-generation sequencing, summarised in Figure 2. In this section, we summarise genome-wide prognostic methylation biomarker discovery studies in primary PCa tumours, with all but one study performed on RP tissue (Table 2). Table 2 highlights the novel biomarkers that were further validated within the original or subsequent studies, and details the top 20 genes for studies that found a large number of significantly associated markers. We refer readers to the original studies for the full lists of methylation markers. We focus on those studies that used measures of disease risk to identify potential biomarkers (for example, comparing methylation of patients of different GS, or different survival outcomes), rather than studies that compared methylation differences between benign and tumour tissue to identify disease-specific biomarkers, with assessment of their prognostic value as only a secondary step [113,114,115,116,117,118,119].

#### 2.2.1. Restriction-Based Methylation Sequencing Studies

The first genome-wide prognostic methylation biomarker discovery study in 2007 used methylation-sensitive arbitrarily primed PCR [139] and methylation CpG island amplification [140] to find markers that could distinguish between patients with high GS (8–10, *n* = 5) vs. those with low GS (2–6 with no grade 4 or 5 patterns; *n* = 5), as well as markers that could predict early BCR following RP (*n* = 5 no BCR (> 4 yrs), *n* = 5 early BCR (< 2 yrs)) [120]. In this approach, methylation-(in)sensitive restriction enzymes were used to digest DNA, and the resulting fragments were screened for methylation differences between patient groups, and sequenced if differences were found. The top 51 markers, along with 11 candidate markers, were validated in two large, independent cohorts, using a custom methylation oligonucleotide microarray (Cohort 1, *n* = 304) and MethyLight qMSP assays (Cohort 2, *n* = 233) [120]. *GRP7*, *ABHD9* and *Chr3*-*EST* were significantly hypermethylated in high GS patients and could distinguish between no BCR and early BCR, independent of patient GS in Cohort 1 [120]. These associations were validated in Cohort 2, where increased methylation of *ABHD9* and *Chr3-EST* correlated with high-grade disease and early BCR, even after adjusting for GS, pathological T-stage and margin status [120]. A subsequent study reported only univariate associations between *ABHD9* methylation and BCR in a larger cohort (*n* = 605) [80], whilst another saw no *ABDH9* methylation difference with BCR status (*n* = 407) [141].

A second restriction enzyme-based method used in genome-wide screening is Enhanced Reduced Representation Bisulphite Sequencing (ERRBS) platform. Similarly, ERRBS involves enzymatic digestion of DNA at CpG sites using *MspI*, followed by size selection and bisulphite sequencing, with the main advantage of enabling single base pair resolution profiling of CpG sites in GC-rich genomic regions such as promoter CpG islands [142,143]. Only one study has used this platform, profiling the methylome of a small discovery set of PCa patients, comparing indolent (localised disease with no recurrence; *n* = 7) vs. advanced cancers (aggressive castration-resistant PCa; *n* = 6) [121]. A series of differentially methylated CpG islands were identified using linear model analysis, and a prognostic panel of 13 hypermethylated CpG islands (see Table 2) was built using random forest classification. This panel successfully discriminated between indolent and advanced cancers in the validation cohort (*n* = 16 indolent, *n* = 8 advanced, MassARRAY EpiTYPER) with an AUC of 0.975 in 10-fold cross validation [121]. Of the genes included in the panel, *GSTP1* has been widely studied as a diagnostic and prognostic biomarker of PCa (see above) [44,45,144], whilst *GRASP* and *TPM4* have been previously shown to be differentially methylated in PCa compared to normal prostate tissue [132].

#### 2.2.2. Capture-Based Methylation Sequencing Studies

Capture-based DNA methylation sequencing for PCa prognostic studies were first used in 2015. The approach involves the capture of methylated sequences by methyl-CpG binding domain (MBD) protein after shearing of genomic DNA, followed by massive parallel sequencing of enriched sequences [145,146]. A limitation of this approach is that it does not provide a single nucleotide resolution, instead identifying regions with multiple methylated CpGs. Only one study has used this method to profile and identify methylation differences between low and high GS tumours (*n* = 6 vs. 9) [122]. They reported hypermethylation of 4932 regions in high-grade disease [122]. Extensive genomic and functional characterisation of these Differentially Methylated Regions (DMRs) were conducted, including comparison with publicly available data from The Cancer Genome Atlas (TCGA) project [147]. This allowed them to validate the association between methylation and high grade disease at 101 DMRs, and show that a subset of these DMRs correlated with gene expression changes that associated with poorer survival in PCa patients, including the *CCDC8* and *HOXD4* gene [122].

#### 2.2.3. DNA Methylation Microarray Studies

Microarrays have become the most popular technology for genome-wide DNA methylation profiling for biomarker discovery. For this method, DNA is first treated to enable later distinction between methylated and unmethylated sites (using methylated DNA immunoprecipitation, methylation-specific restriction enzymes or bisulphite-conversion). The DNA is then hybridised to unique oligonucleotide probes of CpGs on arrays, labelled with a fluorescent dye and imaged, and the signal is used to determine single nucleotide resolution CpG methylation. Several DNA methylation microarrays have been produced, including the Agilent Human CpG Island Microarray [148] (237,220 CpGs) (Figure 2) and Illumina DNA methylation microarrays, with successive array updates interrogating a broader range and number of CpG sites across the genome: GoldenGate Cancer Panel I (1505 CpGs) [149], Infinium HumanMethylation 27K Microarray (HM27K: 27,578 CpGs) [150] and Infinium HumanMethylation 450K Microarray (HM450K: 485,577 CpGs) [151] (Figure 2).

##### Agilent Human CpG Island Microarray

An initial genome-wide study in 2009 using the Agilent Human CpG Island Microarray [123] laid the foundation for a number of subsequent validation studies [124,125,126,127,128], leading to the identification of several robust methylation markers with prognostic value. The original study screened for methylation differences between patients with GS 6 (*n* = 10) and GS 8 (*n* = 10), finding 493 CpG sites (223 individual genes) that could distinguish between the patient groups. A candidate marker, *HOXD3*, was selected and assessed in an independent set of samples (*n* = 20 GS6 vs. *n* = 19 GS8), validating *HOXD3* hypermethylation in GS8 compared to GS6 patients. Further studies investigated *HOXD3* [124] and *GBX2* [126] as individual markers of progression, using BCR as their outcome of interest. *HOXD3* was associated with BCR in univariate analysis (*n* = 147 no BCR, *n* = 85 BCR), and a methylation score combining *HOXD3* with pathological T-stage was found to be an independent predictor of BCR [124]. *GBX2* methylation was assessed in methylation data from TCGA (*n* = 435 no BCR, *n* = 43 BCR), and in a second cohort (*n* = 202 no BCR, *n* = 52 BCR). In both cohorts, associations were observed between *GBX2* methylation and BCR, with *GBX2* methylation shown to have potential as an additive predictor when combined with PSA levels at diagnosis (Cohort 2) [126].

The same research team also investigated *HOXD3* in combination with other markers [125,127]. One study combined *HOXD3*, *TGFß2* (another differentially methylated gene from the original discovery study) and *APC*, an a priori candidate marker [125]. Using a cohort from an earlier study [124] (*n* = 219), they found that this multigene panel improved prediction of BCR over any individual markers, and was independent of existing clinicopathological variables [125]. In another study they used a penalized cox regression method to develop a 4-gene (4-G) prognostic model for BCR, consisting of *APC*, *HOXD3*, *TGFß2* and *CRIP3* [127]. The 4-G model associated with BCR as well as progression to post-surgical therapies (hormone and salvage radiotherapy) in two large cohorts (Cohort 1: *n* = 202 no BCR, *n* = 52 BCR; Cohort 2: *n* = 159, *n* = 40 BCR) [127]. Most recently, the prognostic ability of the 4-G model was investigated in presurgery prostate biopsy specimens (*n* = 61 no BCR, *n* = 25 BCR) [128]. The 4-G methylation panel was able to prognosticate BCR, late recurrence (BCR 5 & 7 yrs post-RP) and eventual progression to postsurgery treatments [127]. Additionally, a study from another team found strong evidence of *HOXD3* hypermethylation in BCR progression (*n* = 303 no BCR, *n* = 104 BCR) [141]. These studies provide strong support for *HOXD3* hypermethylation as a robust marker for BCR progression. Further studies examining the prognostic utility of *HOXD3* and other genes in the panel are warranted, particularly in cohorts with more clinically relevant endpoints for aggressive disease, such as metastatic relapse and PCa death.

##### GoldenGate Cancer Panel I Microarray Platform

Between 2014 and 2016, two research groups published studies using Illumina’s GoldenGate Cancer Panel I Microarray platform to identify novel biomarkers for disease risk. In the first study, a Support Vector Machine was used to build a classification model, generating a signature consisting of 55 probes across 46 genes (including *ALOX12*, *PDGFRB*) [129]. The signature, termed “PHYMA”, was able to distinguish between low and high GS tumours, and a high PHYMA score associated with poorer survival outcome (adjusted for clinical T-stage and GS) (*n* = 87). Trending associations between PHYMA score and GS, but not BCR, was observed in a separate cohort (*n* = 59) [129]. The second study using the GoldenGate Cancer Panel reported that a gene hypermethylation profile based on hierarchical clustering of patients (see Table 2 for details), as well as hypermethylation at individual markers *GSTM2* and *MCLY2*, independently predicted BCR risk. The concurrent methylation of the two markers was also associated with PCa death, however no further validation of the study findings in a separate independent cohort was conducted [130].

##### HM27K Platform

Two studies published in 2011–2012 used the HM27K Microarray platform for prognostic discovery biomarker studies of PCa. One study (*n* = 86) reported increased methylation at 4 CpGs (*KCNK4*, *WDR86*, *OAS2*, *TMEM179*) in patients with a shorter time to BCR, although no further validation of these markers was conducted [131]. Another study performed a number of binary comparisons of methylation levels: no recurrence (*n* = 75) vs. recurrence (BCR or clinical recurrence (local recurrence or metastatic relapse), *n* = 123); BCR (*n* = 43) vs. clinical recurrence (*n* = 80); and local (*n* = 44) vs. metastatic relapse (*n* = 36). The discovery analysis found 75, 16 and 68 genes significantly methylated in each analysis, respectively. Several markers from each group of analyses were assessed by pyrosequencing in 80 patients (*n* = 20 per clinical endpoint), validating many of the nominated candidate genes (see Table 2 for details), including *RUNX3*, a candidate gene previously studied in a priori prognostic biomarker studies [79,82].

##### HM450K Platform

The HM450K extended the HM27K probe design to provide coverage of a more diverse set of genomic categories and regions [150,151]. The platform has been widely used for prognostic biomarker discovery (since 2016) and in the generation of publicly available data, including the TCGA dataset of nearly 500 PCa methylomes, which is commonly used by researchers for biomarker discovery or validation [147]. A study by Geybels et al. used TCGA HM450K methylation data to identify methylation differences between PCa patients with low GS (≤6, *n* = 65) vs. high GS (8–10, *n* = 88) [133]. The elastic net method was used to build a signature consisting of 52 CpG sites across 32 genes, many of which were novel prognostic candidates. The signature was then tested in HM450K data from a larger validation cohort (*n* = 523) for its ability to predict disease progression (any of BCR, metastases and/or PCa death) and was found to be an independent predictor of progression in multivariate analysis including GS, pathological T-stage and diagnostic PSA level in all patients, and in a subset of GS 7 patients [133]. Another study from the same lab used the HM450K data from 430 primary PCa tissues, to identify 42 DNA methylation biomarkers that could predict the more serious endpoint of metastatic-lethal progression [134]. In total, eight of these CpG sites validated in a small validation cohort (*n* = 65, HM450K), with methylation at four of these sites observed to complement GS in discriminating between nonrecurrent and metastatic-lethal patients [134]. A follow-up study by the same research group used pyrosequencing to technically validate five of the eight differentially methylated CpG sites (in *ALKBH5*, *ATP11A*, *FHAD1*, *KLHL8*, *PI15*) [135]. They then used a training cohort (*n* = 392) to build a model, based on the five sites, from which they calculated a prognostic methylation score for prediction of metastatic-lethal progression. In a multivariate model with GS, a four-fold increase in risk of metastatic-lethal progression was reported in the testing cohort (*n* = 34), for each unit increase in the methylation score, and the methylation score outperformed prediction by GS alone [135].

A novel approach by Mundbjerg and colleagues used the HM450K to profile multiple samples per patient (different tumour foci, adjacent normal tissue, lymph node metastases and normal lymph nodes) from a cohort of patients who had undergone RP for multifocal disease (*n* = 14 patients, *n* = 92 samples). They then used a GLMnet algorithm to categorise the aggressiveness of individual PCa foci based on how well they matched the methylation profile of the lymph node metastasis. The resulting aggressiveness classifier consisted of 25 CpG sites (including in *NXPH2*, *TRIB1* and *PCDHA1-PCDHA8*), and was successfully validated in the TCGA cohort (*n* = 351) through accurate prediction of lymph node metastases and invasive pathological stage T3 tumours [136]. Finally, the most recent HM450K study which aligns with our criteria of prognostic discovery, used random-forest-based modelling to identify markers that could differentiate between good prognosis, defined as organ-confined disease (pT2) and no BCR for at least 5 years (*n* = 35), and poor prognosis, defined as systemic metastatic disease with recurrence within 3 years (*n* = 35) [137]. A DNA methylation-based classifier consisting of 598 sites was developed, validating in two independent cohorts of patients with publicly available methylation data, based on the same selection criteria (ICGC cohort *n* = 63, TCGA cohort: *n* = 84) [137]. Further analyses highlighted the independent prognostic value of a gene overlapping one of the 598 sites, with immunostaining analysis reporting significant association between loss of ZIC2 protein expression and poorer prognosis (adjusted for GS, pathological T- stage, nodal stage and PSA) [137].

In summary, technological advances now mean that many hundreds of thousands of CpG sites can be profiled simultaneously, which has provided a more complete view of the complexity and heterogeneity of the PCa methylome. This has enabled the discovery of more accurate and novel biomarkers for PCa prognosis that aid or outperform existing clinicopathological factors. These range from individual markers (e.g., *ABHD9*, *HOXD3*, *GBX2*, *RASGRF2*) to methylation signatures (e.g., 4-G model, PHYMA). However, with an average follow-up of just approximately 5 years across the studies, most focus on short-term clinical endpoints such as BCR. To discover and validate novel DNA methylation biomarkers for the most important clinical endpoints of metastasis and PCa specific mortality, further research needs to be conducted on large independent cohorts with extensive long-term follow-up data (≥15 years) [16]. Furthermore, the genome-wide methods described above are still limited in the number of CpGs assessed (Figure 2), and have a strong bias towards targeting methylation in CG rich regions of gene promoter and CpG islands [145]. More recent techniques, such as Illumina’s EPIC microarray, cover more distal regulatory genomic regions [151], and the ‘gold standard’ Whole Genome Bisulphite Sequencing (WGBS) technique can profile all approximately 28 million CpGs in the methylome [152] (Figure 2). An expanded search of the methylome will enable comprehensive discovery of novel biomarkers for PCa prognosis.

## 3. Non-Invasive Detection of Prognostic DNA Methylation Markers in Liquid Biopsies

There is widespread interest in using ‘liquid biopsies’ as a minimally invasive means to improve the accuracy and safety of cancer diagnosis, risk-stratification and disease monitoring. Liquid biopsies include bodily fluids, such as blood, urine, saliva and cerebrospinal fluid, which can be sampled for the presence of circulating tumour cells, cell-free (cf) DNA (released from tumour cells by apoptosis, necrosis and active secretion) and tumour-secreted exosomes containing RNAs, DNAs and proteins. A liquid biopsy offers the opportunity to gain a more comprehensive profile of the heterogeneous molecular landscape of the tumour at diagnosis and during tumour evolution over the course of the disease and treatment. This is particularly relevant in PCa, as the majority of patients have multifocal disease, meaning that the information from a single tissue biopsy may not reflect the dynamics of all tumour foci in the prostate, which can have variable aggressiveness and progression [136].

DNA methylation biomarkers are particularly pertinent in the liquid biopsy setting. In contrast to the limited number of recurrent genetic mutations in cancer, aberrant DNA methylation events tend to be tissue and cancer-type specific and occur across larger genomic regions, allowing DNA methylation to be easily targeted for measurement [153]. The recent development of new technologies has greatly contributed to the ability to sensitively measure DNA methylation [55]. This is highly relevant in liquid biopsy samples where tumour DNA may be present at very low concentrations i.e., < 0.01% of the total DNA content [154,155]. For example, *GSTP1* hypermethylation, one of the most common epigenetic events in PCa tumour specimens, has been readily detected in liquid biopsy samples from PCa patients, such as urine, semen, blood serum and plasma samples [156].

### 3.1. Urine-Based Methylated Biomarkers

Early studies showed the detection of *GSTP1* methylation in urine from patients with PCa; however diagnostic sensitivity was poor (less than 30%) [157]. Expanding the panel to a three-gene signature (*GSTP1*, *APC* and *RARB*) improved sensitivity to 60% [158]. More recently a six-gene methylation panel has been developed, termed ‘Epigenetic Cancer of the Prostate Test in Urine’ (epiCaPture), which targets *GSTP1*, *SFRP2*, *IGFBP3*, *IGFBP7*, *APC* and *PTGS2* [159]. epiCaPture was applied to urine samples of men with PCa and showed significant associations between DNA methylation and disease aggressiveness, with AUC of 0.64, 0.86 and 0.83 for detecting PCa, high-grade PCa, and high-risk PCa, respectively. Overall, the study concluded that epiCaPture can accurately determine risk compared to two widely used risk stratification systems, D’Amico [26] and CAPRA [160]. In another study, a two-gene methylation panel (*HOXD3* and *GSTP1*) was developed called Prostate Cancer Urinary Epigenetic (ProCUrE) [161]. When applied to urine samples, the positive predictive value of this panel was 59.478¨C%, higher than PSA (38.2–72.1%), for all risk category comparisons. In addition, Moreira-Barbosa et al. assessed methylation of two different gene panels comprising (*miR1−93b*/*miR3−4b*/*c*) and (*APC*, *GSTP1*, *RARB*) in tissue and urine; they showed that a combination of methylation measurements from the two panels in urine independently predicted shorter disease-specific survival [162]. Hypermethylation of the *RASSF1* promoter has also been reported for its prognostic value as a urine-based methylated biomarker [93]. In this study, a multivariate model of *RASSF1* methylation together with pathological T-stage was the most significant predictor of BCR in patients (GS 6) in both tissue and urine samples [93]. Overall, these studies highlight the potential of DNA methylation as a urine-based prognostic biomarker in PCa.

### 3.2. Blood-Based Methylated Prognostic Biomarkers For cfDNA Testing

In cancer patients, a proportion of circulating cfDNA is derived from tumour cells, i.e., circulating tumour DNA (ctDNA). cfDNA can be isolated from blood plasma or serum and is present at very low concentrations, ranging from approximately 0–50 ng/mL in healthy individuals; in cancer patients, the proportion of ctDNA can vary between 0.01% to more that 90% of the cfDNA.

A number of studies have shown that methylated *GSTP1* in circulating cfDNA has prognostic value [163,164,165]; for example, Mahon and colleagues showed that *GSTP1* methylation in cfDNA was associated with overall survival and response to chemotherapy in men with advanced PCa [164]. Importantly, this study demonstrated that *GSTP1* methylation levels prior to and after one chemotherapy cycle were stronger predictors of overall survival than changes in PSA levels at 3 months post-chemotherapy. More recently, Hendriks and colleagues reported hypermethylation of *GSTP1* and *APC* in plasma cfDNA, together with the concentration of cfDNA, to be statistically significant as a prognostic biomarker for overall survival in castration-resistant PCa [166]. Further cfDNA studies report the prognostic utility of *GSTP1* methylation in combination with other frequently methylated genes; for example, *GSTP1* and *RASSF2A* methylation [59] and *GSTP1*, *RASSF1* and *RARB* methylation [167].

Other genes have shown promise as prognostic methylation biomarkers in cfDNA in PCa. Horning and colleagues showed that promoter hypermethylation of *SRD5A 2* and *CYP11A 1* was associated with BCR and poorer prognosis [168]. In another study, cfDNA methylation of the *APC*, *GSTP1*, *RASSFI*, *MDRI* and *PTGS2* genes was associated with overall survival time in men with advanced PCa [60]. Additionally, MSP on a cohort of *n* = 117 patient serum samples showed that *PCDH8* methylation was an independent predictive risk factor for BCR-free survival (*p* < 0.007) in low GS (< 7) PCa patients after surgery [169]. Overall, these studies highlight the potential value of DNA methylation biomarkers in cfDNA as prognostic indicators of relapse.

## 4. Conclusions and Future Directions

Research over the last two decades has shown the potential of DNA methylation as a biomarker for PCa prognosis. DNA methylation biomarker discovery has accelerated rapidly with the emergence of affordable, scalable, whole-genome profiling techniques. However, the ongoing technological advancements are bringing new analytical challenges, such as establishing the best way to aggregate methylation data across genomic regions, control for multiple tests; combine methylation with other ‘omic’ data types and select and prioritise the most important prognostic features from which to build predictive models [170]. A number of new analytical approaches have been made to address these issues (for example, [171,172,173,174,175]) Given the ever-increasing sophistication of technologies, and thus growing number of high-dimensional datasets, bioinformatic method development will continue to be a high-priority research area.

Even with the most sophisticated laboratory and bioinformatic tools for biomarker discovery, the ultimate test of whether a prospective methylation biomarker is prognostic is through validation in multiple, appropriately sized, independent cohorts. One of the obstructions to validation, and therefore translation of new DNA methylation biomarkers to the clinic, is the dearth of suitable, publicly available methylation datasets with adequate clinical follow-up data. Indeed, the flagship TCGA PCa methylation dataset has only short-term follow-up clinical data, and so cannot be used to fully assess the prognostic value of putative methylation biomarkers. For existing methylation datasets such as this, their utility for prognostic research would be increased through the continued collection of long-term follow-up data. 

A notable problem is that studies frequently use the same few public cohorts for discovery and/or validation, which may be leading to biased results across the field. To advance the field, new PCa methylation public datasets need to be generated. An emphasis should be placed on using the very latest laboratory techniques which allow for full genome screening, such as the comprehensive EPIC microarray or WGBS, which will increase the likelihood of identifying novel biomarker regions. Another limitation in this field is the predominant focus on using Caucasian or European ancestry based populations, with only a handful of studies to date investigating non-Caucasian patients [96,176]. More ethnically diverse populations need to be investigated for discovery of population-specific prognostic markers, as well as examining how well promising biomarkers found in Caucasian populations translate across other ethnicities. Many of the highest impact journals now have an open data policy. Going forward, this open data ethos should be adopted by more researchers and publications, as it not only provides new resources for other researchers to use in their validation efforts, but also allows transparency in the research method, which ultimately improves the quality of research in the field overall.

Finally, we have discussed the advances in noninvasive DNA methylation prognostic biomarker research. Looking ahead, clinical translation of this research will be a priority as liquid biopsies offer a number of advantages over tissue-based methods, such as reducing side-effects like infection and surgical complications [17,18], and allowing serial collection of samples during the course of monitoring or therapy to provide opportunities for timely therapeutic interventions [177]. This should be paired with the utilisation of DNA methylation assays suitable for application in clinical settings (for example, Multiplex Bisulfite PCR Sequencing [178,179]) which are cost-effective, scalable, reproducible and capable of sensitively detecting methylated tumour DNA in limited clinical material such as liquid biopsies. In conclusion, DNA methylation shows great potential as a prognostic biomarker and could thus transform the clinical management of PCa patients. Key to the successful implementation of prognostic biomarkers is the ability to apply them in diagnostic samples, such as needle biopsy or liquid biopsy samples. Ultimately, the development of specific guidelines for clinical use still requires extensive validation of the best candidate genes in a range of tissue types in independent cohorts with long-term follow-up, for determination of methylation level cut-offs and prognostic validation.

## Figures and Tables

**Figure 1 cancers-12-02993-f001:**
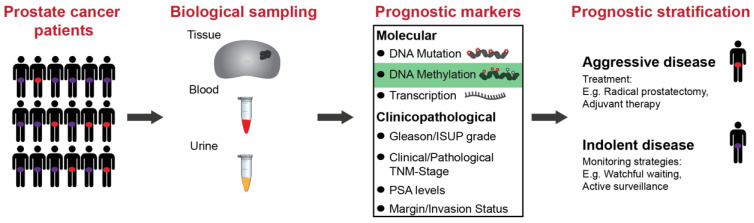
Schematic showing the pathway for the application of biomarkers for improved prognostic stratification in prostate cancer patients. Prostate cancer is a heterogeneous disease and identifying those patients at diagnosis that have aggressive vs. indolent disease is critical in informing the clinical management of these patients. Biological sampling includes tissue biopsies, and blood and urine samples. These are assayed using molecular biomarkers, including DNA mutations, DNA methylation and transcription, and clinicopathological markers, such as GS/ISUP grade and tumour staging. The ultimate goal is to improve the prognostic stratification of patients to inform the optimal treatment strategies for prostate cancer patients. TNM: tumour/nodes/metastasis staging; PSA: prostate-specific antigen.

**Figure 2 cancers-12-02993-f002:**
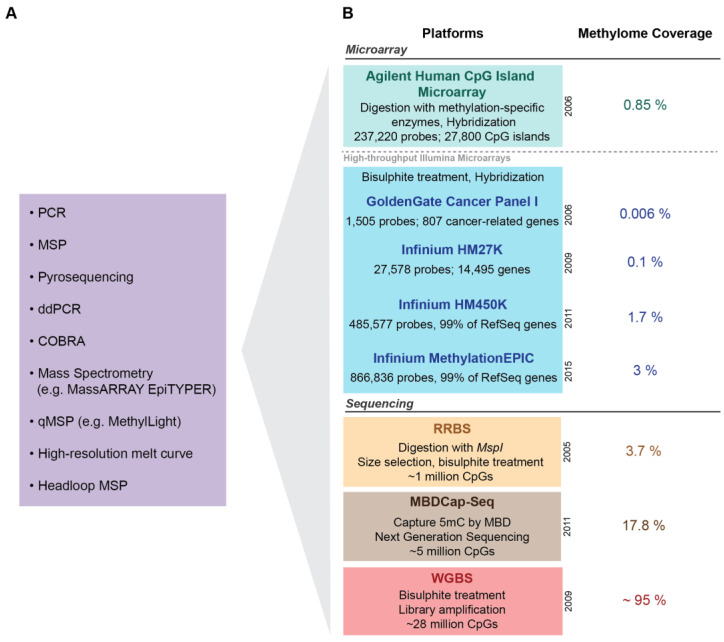
Methylation profiling approaches for biomarker discovery and validation. (**A**) Candidate gene approaches: Targeted candidate gene approaches used in a priori studies and for the validation of novel candidate markers. (**B**) Genome wide approaches: Microarray and sequencing-based genome-wide approaches used for the discovery of novel biomarkers—comparison of the methodology, number of CpGs (cytosine-guanine dinucleotides) and/or genes targeted and coverage of the methylome across the different platforms. RefSeq: Reference Sequences; PCR: polymerase chain reaction; MSP: methylation-specific PCR; ddPCR: droplet digital PCR; COBRA: combined bisulphite restriction analysis; qMSP: quantitative methylation-specific PCR; HM: human methylation; RRBS: reduced representation bisulphite sequencing; MBDCap-Seq: methyl-CpG binding domain capture sequencing; WGBS: whole genome bisulphite sequencing.

**Figure 3 cancers-12-02993-f003:**
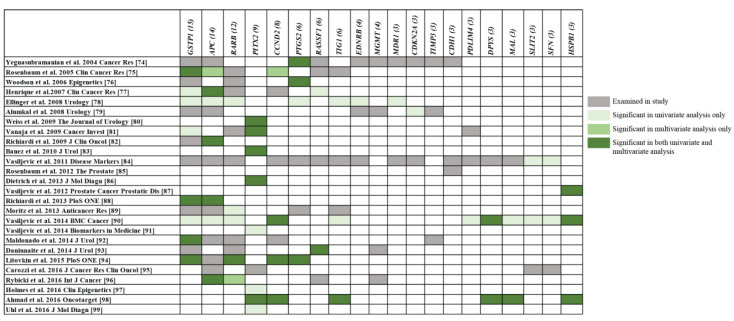
Genes studied in ≥ 3 prognostic candidate marker studies. Each row represents a candidate gene study, and each column represents a gene for which three or more studies have investigated its methylation profile as a prognostic marker of PCa. The number in brackets indicates the number of studies the gene has been examined in. For each of the 26 studies, genes that were investigated have been highlighted: grey = examined in study but no significant associations were found; faded green = significant associations found in univariate analysis only; light green = significant associations observed in multivariate analysis only; green = significant associations reported in both univariate and multivariate analysis.

**Table 1 cancers-12-02993-t001:** Candidate (a priori) prognostic methylated tissue-based biomarker studies.

Study [Ref] ^a^	Primary Outcomes Examined	Total COHORT SIZE	Additional COHORT DETAILS	Follow-Up (Years)	Method	Genes Examined	Genes Validated ^b,c,d,e^	Results ^g^
Yegnasubramanian et al. [74]	BCR	*n* = 36	*n* = ns BCR	Range: 8–13	qMSP	*GSTP1*, *APC*, *RASSF1*, *PTGS2*, *MDR1*, *HIC1*, *EDNRB*, *ESR1*, *CDKN2a*, *CDKN2b*, *p14/QRF*, *MGMT*, *hMLH1*, *TIMP3*, *DAPK1*, *CDH1*	U: *PTGS2* M: *PTGS2* [+ GS + pathological T-stage]	U: HR 2.82 (1.07–7.44), *p* = 0.04 M: HR 4.26 (1.36–13.36), *p* = 0.01
Rosenbaum et al. [75]	Primary: Progression Secondary: Metastatic relapse and/or PCa death	*n* = 74	*n* = 37 no progression, *n* = 37 progression (*n* = 14 BCR, *n* = 16 metastatic relapse, *n* = 7 PCa death) [all GS 7 (3 + 4)]	Min: 7, Median: 9	qMSP	*APC*, *CCND2*, *GSTP1*, *TIG1*, *RASSF1*, *RARB*	Progression: U: *GSTP1* M: [+ age > 60] (1) *GSTP1 + APC*; (2) *GSTP1* + *APC* and/or *CCND2* Metastatic relapse and/or PCa death: U: None M: None	Progression: U: HR 0.34 (0.13–0.88), *p* = 0.03 M: (1) *GSTP1* [HR 0.23 (0.09–0.64), *p* = 0.004], *APC* [HR 3.0 (1.42–6.32), *p* = 0.0004; (2) *GSTP1* [HR 0.29 (0.11–0.77), *p* = 0.01], *APC* or *CCND2* methylated: [HR 1.84 (0.92–3.72), *p* = 0.09], *APC* + *CCND2* [HR 4.33 (1.52–12.33), *p* = 0.01] Metastatic relapse and/or PCa death: U: non-sig; M: non-sig
Woodson et al. [76]	BCR	*n* = 60	*n* = 49 no BCR, *n* = 11 BCR	Mean (SD): No BCR-6.5 (3.2) BCR-4.7 (2.8)	qMSP	*GSTP1*, *RARB*, *CD44*, *PTGS2*	U: (1) *CD44*; (2) *PTGS2* M: *CD44* + *PTGS2* [+ GS]	U: (1) OR 6.83 (1.67–27.99), *p* = 0.008; (2) OR 4.38 (1.13–17.40), *p* = 0.04 M: *CD44* + *PTGS2* [OR 8.87 (1.85–42.56), *p* = 0.006]
Henrique et al. [77]	Primary: PCa death Secondary: BCR	*n* = 83	*n* = 15 PCa death, *n* = 37 BCR [Sextant biopsy cores]	Median (range): 3.7 (0.5–5)	qMSP	*APC*, *CCND2*, *GSTP1*, *RARB*, *RASSF1*	PCa death: U: *APC* M: *APC* [+ Clinical T-stage] BCR: U: (1) *APC*; (2) *GSTP1*; (3) *RASSF1* M: *APC* [+ Clinical T-stage]	PCa death: U: Log-rank *p* = 0.010 M: OR 3.51 (1.23–9.96), *p* = 0.018 BCR: U: Log-rank (1) *p* = 0.002; (2) *p* = 0.047; (3) *p* = 0.019 M: OR 2.58 (1.29–5.16), *p* = 0.008
Ellinger et al. [78]	BCR	*n* = 41	*n* = 28 no BCR, *n* = 13 BCR	Mean; median (range): 2.3; 1.7 (0.5–6)	qMSP	*Annexin2*, *APC*, *EDNRB*, *GSTP1*, *PTGS2*, *MDR1*, *RARB*, *Reprimo*, *TIG1*	U: (1) *APC* + *Reprimo*, (2) > 5 genes hypermethylated together M: None	U: Log-rank (1) *p* = 0.0078; (2) *p* = 0.0074 M: non-sig
Alumkal et al. [79]	BCR	*n* = 151	*n* = 104 no BCR, *n* = 47 BCR	At least 5 years	Nested MSP	*GSTP1*, *MGMT*, *ASC*, *CDKN2A*, *EDNRB*, *CDH13*, *CD44*, *TIMP3*, *RUNX3*, *APC*, *WIF1−*	U: *CDKN2A* M: *CDH13* and/or *ASC* [+ *GS* + extra capsular penetration + seminal vesicle involvement +margin status]	U: OR 0.43 (0.19–0.98), *p* = 0.05 M: *CDH13* [OR 5.51 (1.34–22.67), *p* = 0.02], *CDH13* and/or *ASC* [OR 5.64 (1.47–21.7), *p* = 0.01, sensitivity = 72.3%, specificity = 48%]
Weiss et al. [80]	BCR	*n* = 605	*n* = 540 no BCR, *n* = 65 BCR	Median: 5.5	qMSP	*ABHD9*, *Chr3-EST*, *GPR7*, *HIST2H 2B F*, *CCND2*, *PITX2*	U: (1) *ABHD9*; (2) *Chr3-EST*; (3) *GPR7*; (4) *HIST2H 2B F*; (5) *PITX2* (also in GS7 only and GS8 only patients) M: *PITX2* [+ GS + pathological T-stage]	U: (1) HR 1.9 (1.1–3.1), *p* = 0.02; (2) HR 2.1 (12–3.5), *p* = 0.007; (3) HR 2.3 (1.4–3.9), *p* = 0.002; (4) HR 1.9 (1.1–3.1), *p* = 0.018; (5) HR 3.4 (1.9–6.0), *p* < 0.001, GS7 log-rank *p* = 0.007, GS8 log-rank *p* = 0.023 M: HR 2.5 (1.1–5.8), *p* = 0.032
Vanaja et al. [81]	Recurrence within 5 years	*n* = 64	*n* = 32 no recurrence, *n* = 32 recurrence (*n* = 10 BCR, *n* = 10 local, *n* = 12 metastatic relapse)	Range: 0–5	MassARRAY EpiTYPER	*FLNC*, *EFS*, *ECRG4*, *RARB*, *PITX2*, *GSTP1*, *PDLIM4*, *KCNMA1*	U: (1) *FLNC* (6 CpG units), (2) *GSTP1* (3 CpG units), (3) *PITX2* (1 CpG unit), (4) *EFS* (1 CpG unit) (5) Methylation score: top 11 CpG units from *FLNC*, *EFS*, *PITX2*, *PDLIM4*, *KCNMA1* (also subgroups of patients with local recurrence, metastatic relapse and BCR) M: Methylation score [+p re-op PSA + GPSM prognostic score using weighted sum of GS, PSA, seminal vesicle involvement and marginal status]	U: (1) Sensitivity = 71.43–78.57%/Specificity = 62.52–75.12%; (2) 63.33–76.92%/72.73–81.82%; (3) 66.67%/64.29%; (4) 62.50%/60.02%; (5) 71.12%/71.90%; local recurrence only: 80.32%/81.2%; metastatic relapse only: 72.72%/75.14%; BCR: 60.26%/59.42% M: Sensitivity = 80%, Specificity = 81.2%, AUC 0.852
Richiardi et al. [82]	PCa death	1980s cohort: *n* = 216 1990s cohort: *n* = 243	1980s cohort: *n* = 95 no PCa death, *n* = 121 PCa death 1990s cohort: *n* = 167 no PCa death *n* = 76 PCa death	Median (range): 1980s cohort—3.1 (0–14) 1990s cohort—6.3 (0–14)	MSP	*APC*, *RUNX3*, *GSTP1*	1980s cohort: U: *APC* M: *APC* [+ source of tumour tissue + GS + follow-up duration] 1990s cohort: U: (1) *APC*, (2) *RUNX3* M: [+ source of tumour tissue + GS + follow-up duration] (1) *APC*; (2) *APC* (GS < 8 only); (3) *RUNX3*; (4) *RUNX3* (GS < 8 only)	(*p*-values not specified in this study) 1980s cohort: U: HR 1.46 M: HR 1.42 (0.98–2.07) 1990s cohort: U: (1) HR 1.99; (2) HR 1.74 M: (1) HR 1.57 (0.95–2.62); (2) HR 2.09 (1.02–4.28); (3) HR 1.56 (0.95–2.56); (4) HR 2.40 (1.18–4.91)
Banez et al. [83]	BCR	*n* = 476	*n* = 370 no BCR, *n* = 106 BCR	ns	qMSP	*PITX2*	U: *PITX2* (also in GS7 patients only) M: *PITX2* [+ GS + pathological T-stage + margin status] (also in GS7 patients only)	U: HR 2.99 (1.99–4.48), *p* < 0.001; GS7 only: HR 2.0 (1.2–3.3), *p* = 0.005; M: HR 2.39 (1.45–3.94), *p* < 0.001, C-index = 0.77; GS7 only: HR 1.87 (1.1–3.1), *p* = 0.02
Vasiljevis et al. [84]	Low vs. High GS	*n* = 48	*n* = 9 GS6, *n* = 23 GS7, *n* = 7 GS8, *n* = 9 GS9–10	NA	Pyro sequencing	*RARB*, *GSTP1*, *HIN1*, *APC*, *BCL2*, *CCND2*, *CDH13*, *EGFR5*, *NKX2–5*, *RASSF1*, *DPYS*, *MDR1*, *PTGS2*, *EDNRB*, *MAL*, *PDLIM4*, *HLAa*, *TIG1*, *ESR1*, *SLIT2*, *CDKN2A*, *MCAM*, *SFN*, *THRB*, *CDH1*, *TWIST1*	U: *SFN*, *SLIT2*, *SERPINB5* (pairwise measures) M: Not conducted	U: *SFN* + *SERPINB5*: correctly classified 81% of high GS; *SFN* + *SLIT2*: 62%; *SERPINB5* + *SLIT2*: 62% M: NA
Rosenbaum et al. [85]	Primary: Progression Secondary: Metastatic relapse and/or PCa death	*n* = 95	*n* = 47 no progression, *n* = 48 progression (*n* = 22 BCR only, *n* = 17 metastatic relapse, *n* = 9 PCa death) [all GS 7 (3 + 4)]	All: At least 8 years No progression (Median (range))—10 (8–14) Progression (Median)—8	qMSP	*RBM6*, *MT1G*, *CDH1*, *AIM1*, *KIF1A*, *PAK3*	Progression: U: *AIM1* M: *AIM1* [+ age at diagnosis + lymph node status] Metastatic relapse and/or PCa death: U: None M: None	Progression: U: HR 0.4 (0.18–0.89), *p* = 0.02 M: HR = 0.45 (0.2–1.0), *p* = 0.05 Metastatic relapse and/or PCa death: U: non-sig M: non-sig
Vasiljevic et al. [86]	PCa death	*n* = 349	*n* = 258 no PCa death, *n* = 91 PCa death	Median (max): 9.5 (20)	Pyro sequencing	*HSPB1*	U: *HSPB1* (in all samples and in subgroup of GS < 7) M: *HSPB1* [+ GS + extent of disease (proportion of TURP) + PSA + *HSPB1* × GS interaction term]	U: HR 1.77 (per 50% increase) (1.13–2.79), *p* = 0.02; GS < 7: *p* = 0.028 M: *HSPB1* [HR 1.18 (per 10% increase) (0.98–1.41), *p* = 0.075], *HSPB1 × GS* [0.98 (0.97–0.99), *p* = 0.014]; model with *HSPB1* vs. clinical variables only: Δχ^2^ = 6.673, df = 2, *p* = 0.036
Dietrich et al. [87]	BCR	Training cohort: *n* =157 Testing cohort: *n* = 523	Training cohort: *n* = ns BCR Testing cohort: *n* = 414 no BCR, *n* = 109 BCR (same cohort as Banez et al. [83])	ns	qMSP	*PITX2*	Training cohort: U: *PITX2* M: Not conducted Testing cohort: U: *PITX2*, (2) *PITX2* (subgroup of ≥ 75% tumour content), (3) *PITX2* (subgroup of GS7 and ≥ 75% tumour content) M: (1) *PITX2* [+ GS + pathological T-stage + PSA + surgical margins]; (2) *PITX2* [+ tumour cell content + pathological T-stage]	Training cohort: U: 3.479 (1.2599¨C.614), *p*-value not given M: NA Testing cohort: U: HR 2.614 (1.79–53.807), *p* < 0.001; (2) log-rank *p* < 0.001; (3) log-rank *p* = 0.003 M: (1) HR 1.814 (1.232–2.673), *p* = 0.003; (2) HR 1.889 (1.259–2.832), *p* = 0.002
Richiardi et al. [88]	PCa death	*n* = 157	*n* = 114 no PCa death, *n* = 43 PCa death [Non-neoplastic tissue adjacent to prostate tumour] [nested in the 2 cohorts of Richiardi et al. [82]]	Median (range): 6.8 (0.03–24.1)	qMSP	*APC*, *GSTP1*	U: *(1) APC*, *(2) GSTP1* M: *APC + GSTP1* [+ age at diagnosis + year of diagnosis + source of tumour tissue + methylation in prostate tumour tissue + GS] (also in restricted analyses of first 5 years of follow-up)	U: (1) HR 2.38 (1.23–4.61), *p*-value not given; (2) HR 2.92 (1.49–5.74), *p*-value not given M: *APC + GSTP1* [HR 2.40 (1.15–5.01), *p* = 0.032]; first 5 yrs follow-up: HR 3.29 (1.27–8.52), *p* = 0.019
Moritz et al. [89]	BCR	*n* = 84	*n* = 31 no BCR, *n* = 53 BCR [GS 5–7]	Mean; median (range): 4; 1.8 (0–10.9)	qMSP	*APC*, *GSTP1*, *PTGS2*, *RARB*, *TIG1*	U: *RARB* M: None	U: HR 2.686 (1.147–6.291), *p* = 0.023 M: non-sig
Vasiljevic et al. [90]	PCa death	*n* = 367	*n* = 268 no PCa death, *n* = 99 PCa death [TURP tissues of men who chose not to be treated for at least 6 months following diagnosis—TAPG cohort]	Median (range): 9.5 (0.7–19.6)	Pyro sequencing	*GSTP1*, *APC*, *RARB*, *CCND2*, *SLIT2*, *SFN*, *SERPINB5*, *MAL*, *DPYS*, *TIG1*, *HIN1*, *PDLIM4* and *HSPB1*	U: (1) *GSTP1*; (2) *APC*; (3) *RARB*; (4) *CCND2*; (5) *SLIT2*; (6) *SFN*; (7) *MAL*; (8) *DPYS*; (9) *TIG1*; (10) *HIN1*; (11) *PDLIM4*; and (12) *HSPB1* M: *DPYS* + *HSPB1* + *CCND2* [+ GS, PSA + HSPB1 × GS interaction term]	U: All genes had an HR (per 10% increment) between 1.09 and 1.28, and *p* between 2.9 × 10^−6^ and 0.029. M: *DPYS* [HR 1.13 (1.03–1.25), *p* = 0.012], *HSPB1* [HR 2.39 (1.15–4.97), *p* = 0.019], *CCND2* [HR 0.86 (0.75–0.98), *p* = 0.024], *HSPB1 × GS* [HR 0.89 (0.81–0.97), *p* = 0.012], C-index = 0.83 (vs. 0.74 for GS + PSA only)
Vasiljevic et al. [91]	PCa death	*n* = 135	*n* = 90 no PCa death, *n* = 45 PCa death [all GS ≤ 7] [subset of cohort from Vasiljevic et al. [90]]	No PCa death (Mean)—7.8 PCa death (Median (max))—15.3 (20)	Pyro sequencing	*PITX2*, *WNT5A*, *SPARC*, *EPB4L 1L 3* and *TPM4*	U: *PITX2* (FDR adjustment = 5%) M: not conducted	U: OR 1.56 (per 10% increment) (1.17–2.08), adjusted *p* = 0.005 M: NA
Maldonado et al. [92]	Progression	*n* = 452	*n* = 193 no progression, *n* = 259 progression	Range: 3–11	qMSP	*AIM1*, *APC*, *CCND2*, *GPX3*, *GSTP1*, *MCAM*, *RARB*, *SSBP2*, *TIMP3*	U: *GSTP1* M: *GSTP1* [+ age at surgery + pre-op PSA + positive surgical margins + surgery year + pathological T-stage + GS] (organ confined disease only)	U: Wilcoxon rank sum test *p* = 0.01 M: OR 1.73 (1.00–3.02), *p* = 0.05
Daniunaite et al. [93]	BCR	*n* = 149	*n* = ns BCR	No BCR (Median (range))—3.4 (0.2–5.5)	qMSP	*RARB*, *GSTP1*, *RASSF1*, *MGMT*, *DAPK1*, *p16* and *p14*	U: (1) *RASSF1*; (2) *DAPK*; (3) *RASSF1* +/or *DAPK1* M: *RASSF1* [+pT] (GS6 only)	U: (1) HR 2.27 (1.12–4.63), *p* = 0.019; (2) HR 2.55 (1.11–5.84), *p* = 0.045; (3) HR 2.20 (1.06–4.54), *p* = 0.027 M: HR 5.81 (1.08–31.22), *p* = 0.042
Litovkin et al. [94]	Clinical Failure	Training cohort: *n* = 147 Validation cohort: *n* = 71	Training cohort: *n* = 117 no CF *n* = 30 CF Validation cohort: *n* = 58 no CF *n* = 13 CF [High-risk PCa patients: Clinical stage ≥ T3a, GS 8–10 and/or PSA > 20 ng/mL]	Median (range): Training cohort—6.8 (0.1–12.8) Validation cohort—11.5 (1.4–18.8)	Multiplex qMSP	*APC*, *CCND2*, *GSTP1*, *PTGS2* and *RARB*	Training cohort: U: (1) *GSTP1* (trichotomized); *PTGS2* M: [+ pathological T-stage + GS + pre-op PSA] (1) *GSTP1* (trichotomized); (2) *PTGS2* Validation cohort: U: (1) *GSTP1* (trichotomized); (2) *CCND2*; (3) *RARB* M: [+ pathological T-stage + GS + pre-op PSA] (1) *GSTP1* (trichotomized); (2); (3) *RARB*	Training cohort: U: (1) HR 2.96 (1.38–6.36), *p* = 0.005; (2) HR 0.39 (0.18–0.81), *p* = 0.013 M: (1) HR 3.65 (1.65–8.07), *p* = 0.001, C-index = 0.72 (vs. 0.68 for stage + GS + PSA only); (2) HR 0.21 (0.09–0.50), *p* < 0.001 Validation cohort: U: (1) HR 3.34 (1.38–4.87), *p* = 0.003; (2) HR 0.21 (0.07–0.65), *p* = 0.007; (3) HR 3.45 (1.09–10.87), *p* = 0.035 M: (1)HR 4.27 (1.03–17.72), *p* = 0.046, C-index = 0.80 (vs. 0.75 for stage + GS + PSA only); (2) HR 0.19 (0.05–0.79), *p* = 0.022; (3) HR 3.81 (1.09–13.34), *p* = 0.036
Carozzi et al. [95]	PCa death (within 10 yrs of diagnosis)	*n* = 129	*n* = 91 alive > 10 yrs, *n* = 38 died ≤ 10 years [Needle biopsy specimens]	ns	Pyro sequencing	*APC*, *SFN*, *SERPINB5*, *SLIT2*, *PITX2*, *AR*	U: *SERPINB5 ^f^* M: None	U: 2nd quartile [OR 1.54 (0.56–4.23)], 3rd quartile [HR 2.42 (0.91–6.49); *p* = 0.0474 M: non-sig
Rybicki et al. [96]	BCR	*n* = 353	*n* = 262 no BCR, *n* = 91 BCR (White: *n* = 152 no BCR, *n* = 54 BCR African American: *n* = 110 no BCR, *n* = 37 BCR) [Benign prostate specimens—patients who eventually developed PCa]	Median (range): No BCR-6.3 (1–19) BCR-1.9 (0.2–14)	Nested MSP	*RARB*, *APC*, *CTNND2*, *RASSF1* and *MGMT*	U: APC (White patients) M: (1) *APC* (White patients) [+ age at diagnosis + tumour stage + GS + pre-op PSA, treatment type]*;* (2) *APC* (White patients) [+ no other gene methylated + low PSA at cohort entry + inflammation was present]; (3) *RARB* (African American patients) [+ another gene methylated + absence of inflammation]	U: HR 2.07 (1.15–3.74), *p* = 0.02 M: (1) HR 2.26 (1.23–4.16), *p* = 0.01*;* (2) HR 3.28 (1.33–8.11), *p* = 0.01; (3) HR = 3.80 (1.07–13.53), *p* = 0.04
Holmes et al. [97]	BCR	Cohort 1 (TCGA): *n* = 498 Cohort 2: *n* = 300	Cohort 1: *n* = ns BCR Cohort 2: *n* = ns BCR	Mean; median: Cohort 1—1.83; 1.3 Cohort 2—5.5; 5.2	Cohort 1—HM450K Microarray Cohort 2—qMSP	*PITX3*, *PITX2*	Cohort 1: U: (1) *PITX3*; (2) *PITX3* + *PITX2* M: Not conducted Cohort 2: U: (1) *PITX3*; (2) *PITX2*; (3) *PITX3* + *PITX2* M: Not conducted	Cohort 1 U: (1) HR 1.83 (1.07–3.11), *p* = 0.027; (2) HR 2.20 (1.25–3.87), *p* = 0.006; (3) LR = 12.70, log-rank *p* = 0.002 M: NA Cohort 2: U: (1) HR 2.56 (1.44–4.54), *p* = 0.001; (2) see Reference 99; (3) LR = 12.14, log-rank *p* = 0.002 M: NA
Ahmad et al. [98]	PCa death	*n* = 385	*n* = 328 no PCa death, *n* = 57 PCa death [low (0–2) to intermediate (3–5) risk CAPRA scores] [from TAPG cohort in Vasiljevic et al. [90]]	Median (IQR): 11.36 (6.20–14.72)	Pyro sequencing	*HSPB1*, *CCND2*, *TIG1*, *DPYS*, *PITX2*, *MAL*	Methylation score: HSPB1 + CCND2 + TIG1 + NPYS + PITX2 + MAL + CCND2 · HSPB1 interaction term. U: Methylation score M: Methylation score [+ CAPRA]	U: HR 2.72 (1.933.8), *p* < 10^−8^ M: HR 2.02 (1.402.92), *p* < 10^−3^, C-index of full model = 0.74 Sensitivity = 83%, Specificity = 44% (vs. CAPRA only: 68%/44%) At 10 yr follow-up: AUC = 0.74 (vs. CAPRA only: 0.62)
Uhl et al. [99]	Cohort 1: BCR Cohort 2: ISUP grade group (as surrogate for survival)	*n* = 206	Cohort 1: *n* = 208 no BCR, *n* = 52 BCR [same cohort as Cohort 2 in Holmes et al. [97]] Cohort 2: *n* = 32 [core needle biopsy specimens]	Cohort 1 (Mean; median (range))—5.5; 5.2 (0–12.1) Cohort 2—NA	qMSP	*PITX2*	Cohort 1: U: *PITX2* M: None Cohort 2: U: *PITX2* (median, mean and maximum levels) M: Not conducted	Cohort 1: U: HR 1.77 (1.01–3.10), *p* = 0.046 M: non-sig Cohort 2: U: median [*r* = 0.456, *p* = 0.010]; mean [*r* = 0.478, *p* = 0.007]; maximum [*r* = 0.495; *p* = 0.005] M: NA

Abbreviations: AUC = area under the curve; CF = clinical failure; df = degrees of freedom; GS = Gleason Score; HR = hazard ratio; IQR = interquartile range; LR = likelihood ratio; M = multivariate analysis; MSP = methylation-specific PCR; NA = not applicable; non-sig = nonsignificant; ns = not specified; OR = odds ratio; PCa = prostate cancer; PSA = prostate-specific antigen; qMSP = quantitative methylation-specific PCR; r = correlation coefficient; U = univariate analysis. Definitions: BCR: Biochemical recurrence—PSA elevations ≥ 0.2 ng/mL post-RP, except [77] >0.4 ng/mL and [89] >0.1 ng/mL; clinical failure: either of local recurrence or metastatic relapse; clinical T-stage: tumour staging based on results of digital rectal examination, PSA levels and GS. Local recurrence: cancer observed on prostatic bed, confirmed by histological analysis of biopsies; metastatic relapse: metastatic deposits (visceral, bony metastasis) confirmed by positive biopsies or cT/bone scans; pathological T-stage: tumour staging based on pathological examination of surgically removed prostate tissue; PCa death: prostate cancer-specific death; progression: either of BCR, metastatic relapse or PCa death; recurrence: either of BCR, local recurrence or metastatic relapse. ^a^ All studies are on prostate cancer tissues from radical prostatectomy, unless specified. ^b^ Univariate (U) or Multivariate (M) analyses. ^c^ Plus (+) sign indicates variables in the same multivariate model or methylation score together. ^d^ Square bracket ([[]) indicate the clinicopathological factors adjusted for in each multivariate model. ^e^ The use of bracketed numbers; e.g., (1), (2), indicates different genes, sets of genes or multivariate models validated in the respective study. ^f^ Genes validated may have been impacted by high number of missing cases. ^g^ Number in brackets following HR or OR indicate the 95% confidence interval.

**Table 2 cancers-12-02993-t002:** Candidate (a priori) prognostic methylated tissue-based biomarker studies.

Study [Ref] ^a^	Approach (Discovery)	Cohort Size (Discovery)	Follow-Up (Discovery) [Years]	Approach (Validation)	Cohort Size (Validation)	Method (Validation)	Follow-Up (Validation) [Years]	Final Markers Identified (Validation) ^c,d,e,f^	Key Results (Validation) ^g^
**Restriction-based methylation sequencing platforms ^h^**									
1. Methylation sensitive arbitrarily primed PCR and methylated CpG island amplification									
Cottrell et al. [120]	Low GS (2–6 with no grade 4 or 5 patterns) vs. High GS (8–10) Early BCR (< 2 yrs post-RP) vs. no early BCR (> 4 yrs)	*n* = 5 Low GS, *n* = 5 High GS, *n* = 5 no early BCR, *n* = 5 early BCR	Range: 2–4	Low GS vs. High GS Early BCR vs. no early BCR	Cohort 1: *n* = 304 (130 low GS, 96 high GS; 88 no BCR, 63 BCR) Cohort 2: *n* = 233 (28 low GS, 27 high GS; 134 no BCR, 59 BCR)	1. Custom methylation oligonucleotide microarray 2. MethyLight (qMSP) ^b^	Range: 2–4	Low GS vs. High GS: Cohort 1—U: (1) *ABDH9*, (2) *Chr3-EST*, (3) *GPR7*, (4) *NOTCH*, (5) *KBTBD6*; M: not conducted Cohort 2—U: (1) *ABDH9*, (2) *Chr3-EST*; M: not conducted Early BCR vs. no early BCR: Cohort 1—U: (1) *ABDH9*, (2) *ABDH9* (intermediate GS 6, 7 patients only) (3) *Chr3-EST*, (4) *Chr3-EST* (intermediate GS), (5) *GPR7*, (6) *GPR7* (intermediate GS); M: not conducted Cohort 2—U: (1) *ABDH9*, (2) *ABDH9* (intermediate GS), (3) *Chr3-EST*, (4) *Chr3-EST* (intermediate GS); M: (1) *ABDH9* [ + GS + pathological T-stage + margin status], (2) *Chr3-EST* [ + GS + pathological T-stage + margin status]	Low GS vs. High GS: Cohort 1—U: AUC (all Wilcoxon’s *p* < 0.001) (1) 0.71, (2) 0.70, (3) 0.72, (4) 0.71, (5) 0.71; M: NA Cohort 2—U: AUC (all Wilcoxon’s *p* < 0.001) (1) 0.77, (2) 0.79; M: NA Early BCR vs. no early BCR: Cohort 1—U: AUC (Wilcoxon’s) (1) 0.71 (*p* = 0.002), (2) 0.63 (*p* = 0.072), (3) 0.66 (*p* = 0.05), (4) 0.72 (*p* = 0.002), (5) 0.72 (*p* = 0.0002), (6) 0.70 (*p* = 0.005); M: NA Cohort 2—U: AUC (Wilcoxon’s) (1) 0.65 (*p* < 0.001), (2) 0.66 (*p* < 0.01), (3) 0.67 (*p* < 0.001), (4) 0.67 (*p* < 0.01); M: Logistic regression (1) *p* = 0.016; (2) *p* = 0.043, AUC = 0.81 & 0.79 (vs. 0.75 of GS + stage + margin status only)
2. Enhanced Reduced Representation Bisulphite Sequencing									
Lin et al. [121]	Indolent (localised disease, no recurrence) vs. Advanced (aggressive CRPC) PCa	*n* = 7 indolent, *n* = 6 advanced	Indolent (range) 5–6 years	Indolent vs. aggressive PCa	*n* = 16 indolent, *n* = 8 advanced	MassARRAY EpiTYPER	Indolent 3–7 years	Panel of 13 CpG islands: *KCNC2*, *ZDHHC1*, *TBX1*, *CAPG*, *RARRES2*, *GRASP*, *SAC3D1*, *TPM4*, *GSTP1*, *NKX2–1*, *FAM107A*, *SLC13A3*, *FILIP1L* U: Panel M: Not conducted	U: AUC = 0.975 (Sensitivity = 95%; Specificity = 95%) M: NA
**Capture-based methylation sequencing platforms**									
MBD (methyl-CpG binding domain)-isolated genome sequencing (MiGS)									
Bhasin et al. [122]	Low GS (6) vs. High GS (8–10)	*n* = 6 Low GS, *n* = 9 High GS	NA	Low GS vs. High GS	*n* = 46 Low GS, *n* = 203 High GS (TCGA)	HM450K Microarray	NA	U: 101 DMRs including at *CD14*, *PCDHGA11*, *EYA1*, *CCDC8*, *HOXC4*; M: not conducted	U: LIMMA *p* = 2.81 × 10^28^–0.05 (range) M: NA
**Microarray-based platforms**									
1. Agilent Human CpG Island Microarray									
Kron et al. [123]	Low GS (6 (3 + 3)) vs. High GS (8 (4 + 4))	*n* = 10 Low GS, *n* = 10 High GS	NA	1. Low GS vs. High GS	*n* = 20 low GS vs. *n* = 19 high GS (MethyLight)	MethyLight	NA	U: *HOXD3* (detected in *n* = 2 GS6 vs. *n* = 6 GS8); M: not conducted	Sample size too low for statistical U: 17.3% difference in methylation M: NA
				2. Kron et al. [124] GS ≤ 6 vs. GS7 BCR	*n* = 232 (*n* = 101 GS ≤ 6, *n* = 107 GS 7, *n* = 147 no BCR, *n* = 85 BCR)	MethyLight	Mean (range): 4.4 (0.2–9.5)	GS ≤ 6 vs. GS7-U: *HOXD3*; M: not conducted BCR-U: *HOXD3*; M: *HOXD3* x pathological T-stage interaction term [+ GS + pathological T-stage + margin status]	GS ≤ 6 vs. GS7-U: 10.1% difference in av. PMR values, Mann-Whitney U test *p* < 0.001; M: NA BCR-U: Log-rank *p* = 0.043; M: HOXD3 x pT3a [HR 3.78 (1.09–13.17), *p* = 0.037], HOXD3 x pT3b/pT4 [HR 5.23 (1.31–20.96), *p* = 0.019]
				3. Liu et al. [125] GS≤6 vs. GS7 BCR	*n* = 219 (*n* = 138 GS ≤6, *n* = 98 GS 7, *n* = ns BCR) (reduced cohort from Kron et al. [124])	MethyLight	ns	GS ≤ 6 vs. GS7-U: (1) *APC*, (2) *TGFβ2;* M: not conducted BCR-U: (1) *APC*, (2) *HOXD3* + *TGFβ2* + *APC*; M: (1) *HOXD3* + *TGFβ2* + *APC* [+ pathological T-stage + GS (7 and ≥8 groups)], (2) *HOXD3* + *TGFβ2* + *APC* [+ pathological T-stage + GS (7 (3 + 4), 7 (4 + 3) and ≥ 8 groups)]	GS ≤ 6 vs. GS7-U: Mann-Whitney U test (1) *p* = 0.018, (2) *p* = 0.028; M: NA BCR-U: Log-rank (1) *p* = 0.028, (2) *p* < 0.001; M: (1) HR 2.01 (1.14–3.57), *p* = 0.017, (2) HR 2.068 (1.155–3.704), *p* = 0.014
				4. Jeyapala et al. [126] BCR	Cohort 1: *n* = 435, *n* = 43 BCR	Cohort 1: HM450K Microarray (TCGA)	Mean (range): Cohort 1: 1.9 (0–12.6)	Cohort 1—U: *GBX2*; M: not conducted	Cohort 1—U: Mann-Whitney Test cg09094860 [*p* = 0.003], cg00302494 [*p* = 0.01]; M: not conducted
					Cohort 2: *n* = 254 (*n* = 202, *n* = 52 BCR, *n* = 58 IDC/C-positive, *n* = 196 IDC/C-negative)	Cohort 2: MethyLight	Cohort 2: 5.7 (0.1–12.3)	Cohort 2—U: *GBX2* (and in IDC/C-negative patients only); M: (1) *GBX2* [+ GS + Pathological T-stage + pre-op PSA], 2) *GBX2* [+ pre-op PSA]	Cohort 2—U: Mann-Whitney Test *p* = 0.0001, IDC/C-negative patients: Log-rank *p* = 0.002; M: (1) HR 1.02 (1.006–1.034), *p* = 0.004, (2) C-index 0.78 (vs. 0.71 for PSA alone)
				5. Jeyapala et al. [127] BCR salvage RT/hormone therapy	Cohort 1: *n* = 254 (*n* = 202 no BCR, *n* = 52 BCR, *n* = 205 no salvage RT, *n* = 42 salvage RT, *n* = 226 no hormone therapy, *n* = 21 hormone therapy) Cohort 2: *n* = 199 (*n* = 159 no BCR, *n* = 40 BCR, *n* = 180 no salvage RT, *n* = 19 salvage RT, *n* = 177 no hormone therapy, *n* = 22 hormone therapy)	MethyLight	Median (range): Cohort 1: 6.7 (0.1–12.8); Cohort 2: 6.7 (0.2–18.6)	4-G model: *HOXD3*, *TGFβ2*, *CRIP3*, *APC* (candidate) Cohort 1 BCR-U: 4-G model; M: Integrative model = 4-G model [+ CAPRA-S] Salvage RT/hormone therapy: U: 4-G model; M: Integrative model Cohort 2 BCR-U: 4-G model; M: Integrative model Salvage RT/hormone therapy: U: 4-G model; M: Integrative model	Cohort 1 BCR-U: HR 2.72 (1.77–4.17), *p* < 0.001, Sensitivity = 90.9%, Specificity = 35.2%, AUC = 0.740; M: HR 1.49 (1.12–1.99), *p* = 0.006, Sensitivity = 92.9%, Specificity = 43.4%, AUC = 0.846 Salvage RT/hormone therapy: U: HR 2.20 (1.48–3.29), *p* < 0.001;M: HR 1.34 (1.03–1.75), *p* = 0.027 Cohort 2 BCR-U: HR 2.48 (1.59–3.86), *p* < 0.001, Sensitivity = 95.0%, Specificity = 27.5%, AUC = 0.670; M: HR 1.62 (1.17–2.24), *p* = 0.004, Sensitivity = 89.5%, Specificity = 37.3%, AUC = 0.726 (vs. 0.698 for CAPRA-S alone) Salvage RT/hormone therapy-U: HR 1.97 (1.21–3.21), *p* < 0.001, Sensitivity = 91.2%, Specificity = 27.4%, AUC = 0.636; M: HR 1.17 (0.79–1.72), *p* = 0.441, AUC = 0.731 (vs. 0.723 for CAPRA-S alone)
				6. Savio et al. [128] BCR; Late BCR (5 and 7 yrs post-RP) salvage RT/hormone therapy [Biopsy specimens pre-RP]	*n* = 86 (*n* = 61 no BCR, *n* = 25 BCR, *n* = 75 no salvage RT, *n* = 11 salvage RT, *n* = 70 no hormone therapy, *n* = 15 hormone therapy)	MethyLight	Median (range): 5.1 (0.1–16)	BCR-U: none; M: 4-G model [+ pre-op PSA] Late BCR-U: none; M: (1) 4-G model [+ pre-op PSA] (5 yrs), (2) 4-G model [+ pre-op PSA] (7 yrs) Salvage RT/hormone therapy: U: 4-G model; M: (1) 4-G model [+ pre-op PSA], (2) 4-G model [+ CAPRA]	BCR: U: non-sig; M: Sensitivity = 78.6%, Specificity = 64.7%, AUC = 0.714 Late BCR: U: non-sig; M: (1) Sensitivity = 80%, Specificity = 60.5%, AUC = 0.705 (vs. 0.667 for PSA alone), (2) Sensitivity = 76.9%, Specificity = 62.9%, AUC = 0.688 (vs. 0.6 for PSA alone) Salvage RT/hormone therapy-U: Sensitivity = 66.7%, Specificity = 75%, AUC = 0.699; M: (1) Sensitivity = 75%, Specificity = 61.1%, AUC = 0.699, (2) Sensitivity = 76.9%, Specificity = 58.3%, AUC = 0.797
2. Illumina GoldenGate Methylation Microarrays									
Goh et al. [129]	Low GS (6) vs. High GS (8–10) Overall survival	*n* = 87 (*n* = 19 GS6, *n* = 48 GS8–10, *n* = ns death)	Median (range): 4 (0–11.8)	GS (6–9) BCR	*n* = 59 (*n* = 23 GS 6, *n* = 22 GS 7, *n* = 13 GS 8–10, *n* = 18 for BCR analysis)	GoldenGate	No BCR-5 (1–13)	“PHYMA” signature: 55 probes targeting CpG loci of 46 genes, including at *ALOX12*, *PDGFRB* which were selected for functional validation GS: U: PHYMA (GS 6–8) M: not conducted BCR: U: none M: not conducted	GS: U: Logistic regression β-coefficient = 2.28, *p* = 0.2 (trend) M: NA BCR: U: non-sig M: NA
Angulo et al. 2016 Urol Int [130]	• BCR • PCa death	*n* = 26 no BCR, *n* = 32 BCR	Mean ± SD (range): 6.3 ± 3 (0.8–13.8)	No validation	NA	NA	NA	Discovery only: BCR: U: (1) Gene hypermethylation profile of cluster 3 patients (including at *GSTM2*, *GSTP1*, *RARB*, *ALOX12*, *APC*, *PDGFRB*, *SCGB3A1*, *CFTR*, *MT1A*, *PENK*, *NEU1*, *CCNA1*, *MET*, *KLK10*, *RARA*, *MFAP4*, *TERT*, *TBX1*, *TAL*, *MYCL2*, (2) *MT1A*, (3) *ALOX12*, (4) *GSTM2*, (5) *APC*, (6) *MYCL2*, (7) *RARB*, (8) *GSTM2* + *MCLY2* M: (1) Gene hypermethylation profile [+ D’Amico classification]; (2) *GSTM2* [+ ns]; (3) *MCLY2* [+ ns] PCa Death: U: *GSTM2* + *MYCL2* M: not conducted	Discovery only: BCR-U: (1) Log-rank *p* = 0.0054, Cluster 3 vs. 1 [HR 8.4 (1.86–38.46), 3 vs. 2 [HR 2.69 (1.13–5.95), 3 vs. 4 [HR 2.26 (0.89–5.72)]; (2) HR 2.14 (1.06–4.33), log-rank *p* = 0.029; (3) HR 2.21 (1.06–4.55), log-rank *p* = 0.025; (4) HR 4.59 (1.38–15.15), log-rank *p* = 0.0062; (5) HR 1.96 (0.97–3.97), log-rank *p* = 0.05; (6) HR 3.58 (1.6–8), log-rank *p* = 0.0009; (7) HR 2.5 (1.21–5.18), log-rank *p* = 0.01; (8) log-rank *p* = 0.0009; M: (1) Cox regression *p* = 0.064, Cluster 3 vs. 1 [HR 4.37 (0.94–20.41], 3 vs. 2 [HR 2.56 (1.11–5.88)], 3 vs. 4 [HR 2.26 (0.89–5.72)], C-index = 0.708 (vs. 0.679 for D’Amico alone); (2) HR 3.789 (1.11–12.83), *p* = 0.03;(3) HR 2.71 (1.21–6.09), *p* = 0.016 PCa death-U: HR 10.82 (1.96–59.67); log-rank *p* = 0.006; M: NA
3. Illumina Infinium HumanMethylation 27K Microarray									
Kobayashi et al. [131]	BCR	*n* = 86	Range: 0–5.5	No validation	NA	NA	NA	Discovery only: U: *KCNK4*, *WDR86*, *OAS2*, *TMEM179* (FDR ≤ 1%; hypermethylated) M: not conducted	Discovery only: U: ns M:
Mahapatra et al. [132]	Indolent vs. aggressive disease: No recurrence vs. recurrence BCR vs. clinical recurrence Local recurrence vs. metastatic relapse	*n* = 75 no recurrence, *n* = 123 recurrence *n* = 43 BCR, *n* = 80 clinical recurrence *n* = 44 local recurrence, *n* = 36 metastatic relapse	(Mean ± SD): No recurrence 6.2 ± 1.5 BCR: 5.9 ± 1.4 Local recurrence: 4.2 ± 1.7 Metastatic relapse: 4.4 ± 4.0	Indolent vs. aggressive disease: No recurrence vs. recurrence BCR vs. clinical recurrence Local recurrence vs. metastatic relapse	*n* = 20 no recurrence, *n* = 20 BCR, *n* = 20 local recurrence, *n* = 20 metastatic relapse	Pyro sequencing	(Mean ± SD) No recurrence-6.5 ± 1.6 BCR-5.45 ± 1.3 Local recurrence-4.5 ± 1.9 Metastatic relapse-3.5 ± 1.9	No recurrence vs. recurrence: U: (1) *CRIP1*; (2) *RUNX3*; (3) *HS3ST2*; (4) *FLNC*; (5) *RASGRF2* M: Not conducted BCR vs. clinical recurrence: U: (1) *PHILDA3*; (2) *TNFRSF10D*; (3) *RASGRF2* M: Not conducted Local vs. metastatic relapse: U: (1) *BCL11B*; (2) *POU3F3*; (3) *RASGRF2* M: Not conducted	No recurrence vs. recurrence: U: Sensitivity/Specificity, AUC, t-test (1) 65.0%/65.6%, 0.727, *p* = 0.0139; (2) 70.4%/75.3%, 0.788, *p* = 0.0018; (3) 65.0%/60.0%, 0.773, *p* = 0.0115; (4) 70.3%/60.4%, 0.660, *p* = 0.0835; (5) 75.7%/55.2%, 0.682, *p* = 0.0515 M: NA BCR vs. clinical recurrence: U: Sensitivity/Specificity, AUC, t-test (1) 65.6%/65.0%, 0.73, *p* = 0.0129; (2) 60.8%/75.6%, 0.692, *p* = 0.0373; (3) 75.4%/60.3%, 0.761, *p* = 0.0047 M: NA Local recurrence vs. metastatic relapse: U: Sensitivity/Specificity, AUC, t-test (1) 75.2%/60.0%, 0.741, *p* = 0.0091; (2) 65.5%/70.7%, 0.701, *p* = 0.0295; (3) 70.6%/75.4%, 0.748, *p* = 0.0071; M: NA
4. Illumina Infinium HumanMethylation 450K (HM450K) Microarray									
Geybels et al. [133]	Low GS (≤6) vs. High GS (8–10)	*n* = 65 Low GS, *n* = 88 High GS (TCGA)	NA	Progression	*n* = 323 no progression, *n* = 108 progression	HM450K Microarray	Mean (SD): 8.0 (4.2) years	Signature consisting of 52 CpG sites (32 unique genes) including at *RRM2*, *VWA3B*, *MFSD9*, *ANO7*, *GALNTL2*, *SEMA3F*, *ATXN7*, *SLC15A2*, *MME*, *USP17*, *KIAA0922*, *FOXI1*, *URGCP*, *PTPRN2*, *RP1*, *MRPS28*, *MKI67*, *CPT1A*, *KCNMB4*, *TMEM132D*. U: (1) Signature; (2) Signature (GS7 only); (3) Signature (GS7 (3 + 4) only) M: [+ GS+ pathological T-stage + pre-op PSA] (1) Signature; (2) Signature (GS7 only); (3) Signature (GS7 (3 + 4) only)	U: (1) HR 1.78 (per 25% increase) (1.48–2.16), *p* = 2.05 × 10^−9^; (2) HR 1.81 (1.42–2.31), *p* = 1.38 × 10^−3^; (3) HR 1.83 (1.36–2.45), *p* = 5.64 × 10^−5^ M: (1) HR 1.48 (1.21–1.81), *p* = 1.38 × 10^–4^, AUC = 0.78 (vs. 0.73 for GS + pathological T-stage + pre-op PSA only); (2) HR 1.59 (1.24–2.05), *p* = 1.38 × 10^–6^, AUC = 0.76 (vs. 0.64); (3) HR 1.65 (121–2.25), *p* = 1.54 × 10^−3^, AUC = 0.70 (vs. 0.62)
Zhao et al. [134]	Metastatic-lethal progression	*n* = 304 no progression vs. *n* = 24 metastatic-lethal	Mean: Metastatic relapse-8.1 Survival-12.2	1. Metastatic-lethal progression	*n* = 41 no progression, *n* = 24 metastatic-lethal	HM450K Microarray	Mean: 9	U: (1) *ALKBH5*; (2) *ATP11A*; (3) *FHAD1*; (4) *KLHL8*; (5) *PI15*; (6) Intergenic region (chr1); (7) Intergenic (chr16); (8) Intergenic (chr17) M: [ + GS] (1) *ALKBH5*; (2) *FHAD1*; (3) *KLHL8*; (4) *PI15*	U: Mean β difference (*t*-test), AUC, pAUC (1)—5% (*p* = 0.037), 0.66 (*p* = 0.035), 0.001 (*p* = 0.566); (2)—6% (*p* = 0.049), 0.66 (*p* = 0.03), 0.009 (*p* = 0.022); (3)—6% (*p* = 0.007), 0.71 (*p* = 0.003), 0.004 (*p* = 0.159); (4)—10% (*p* = 0.002), 0.75 (*p* = 0.0004), 0.002 (*p* = 0.359); (5)—7% (*p* = 0.029), 0.68 (*p* = 0.014), 0.006 (*p* = 0.074); M: AUC, pAUC, *p* (from likelihood ratio test comparing with model with GS alone: AUC = 0.816, pAUC = 0.010) (1) 0.87, 0.024, *p* = 0.030; (2) 0.86, 0.013, *p* = 0.038; (3) 0.89, 0.008, *p* = 0.014; (4) 0.89, 0.006, *p* = 0.026
				2. Zhao et al. [135] Metastatic-lethal progression	Training dataset (from Zhao et al. [134]): *n* = 344 no recurrence, *n* = 48 metastatic-lethal; Testing dataset: *n* = 11 no recurrence, 23 metastatic-lethal	Pyro sequencing	Training (Mean (minimum))—8 (5) Testing At least 5 years	Methylation score: *ALKBH5* + *ATP11A* + *FHAD1* + *KLHL8* + *PI15* [+ GS] Training-U: not conducted; M: Methylation score Testing-U: not conducted; M: Methylation score	Training-U: NA; M: Logistic regression β-coefficient-ALKBH5 [−0.75], ATP11A [−0.7], FHAD1 [−9.72], KLHL8 [−0.33], PI15 [0.70], GS [1.13] Testing-U: NA; M: Mean difference = 2.49 (*p* = 6.8 × 10^–6^), OR 4.0 (1.8–14.3), *p* = 0.006, AUC = 0.91 (vs. 0.87 for GS alone), pAUC = 0.037 (vs. 0.025 for GS alone), sensitivity at 95% specificity= 74% (vs. 53% for GS alone)
Mundbjerg et al. [136]	Aggressiveness (individual PCa foci vs. matched lymph node metastasis)	*n* = 14 (*n* = 92 samples: multiple tumour foci, adjacent normal tissue, lymph node metastases and normal lymph nodes)	NA	Aggressive (lymph node metastases and pathological stage T3 tumours) vs. non-aggressive	*n* = 351 (TCGA)	HM450K Microarray	Mean: 3.2 years	Aggressiveness classifier: 25 probes, including in *NXPH2*, *NCAPH*, *TRIB1*, *PCDHA1-PCDHA8*, *C3orf37*, *C9orf3*, *CPN1*, *TCF7L2*, *ROBO1*, *GFPT2*, *FBXQ47*, *SKI*, *HDAC9*, *CARS*, *SLC6A17*, *BCAT1*, *GAS1*, *RAI1* U: Aggressiveness classifier M: Not conducted	U: Specificity = 97.4%, Sensitivity = 96.2%, Negative predictive value = 76%, Positive predictive value = 99.7%, Lymph node metastases [Fishers exact test *p* = 9.2 × 10^−5^], pathological stage T3 [Fishers exact test *p* = 2.2 × 10^−7^] M: NA
Toth et al. [137]	Good prognosis vs. Poor prognosis	*n* = 35 good prognosis, *n* = 35 poor prognosis (80% training set, 20% testing set)	Range: 3–5	Good prognosis vs. poor prognosis (Cohort 1, 2) BCR (Cohort 1, 2, 3)	Cohort 1: *n* = 222 (*n* = 63 for prognosis analysis, *n* = ns BCR) (ICGC cohort [138]) Cohort 2: *n* = 477 (*n* = 27 good prognosis, 57 poor prognosis, *n* = ns BCR) (TCGA) Cohort 3: *n* = 12,581 (*n* = 3612 BCR) (for ZIC2 immunostaining analysis only)	HM450K Microarray	Cohort 1 & 2 At least 5 years Cohort 3 (Median (range)) 4 (0.08–20.08)	Signature consisting of 598 CpG sites. Top 20: *CCT8L2*, *NOP56*, *FCRL1*, *OR5W2*, *ZFP36L2*, *PRMT8*, *SLC1A6*, *DOK5*, *CCT8L2*, *ZFP36L2*, *MMP16*, *ESR1*, *ZIC2*, *GPR137B*, *NANOS1*, *LCE3A*, *C11orf87*, *PEG3*, *ZIM2*, *CTSC* Good prognosis vs. good prognosis: Cohort 1—U: (1) Signature, M: not conducted Cohort 2—U: Signature; M: not conducted BCR: Cohort 1—U: Signature, M: not conducted Cohort 2—U: Signature; M: Signature [+ GS] Cohort 3—U: ZIC2 protein; M: ZIC2 protein [+ GS + pathological T-stage + nodal stage + PSA]	Good vs. poor prognosis: Cohort 1—U: AUC = 0.997; M: NA Cohort 2—U: AUC = 0.775; M: NA BCR: Cohort 1—U: Log-rank *p* < 0.0001, M: NA Cohort 2—U: Log-rank *p* < 0.0001; M: Cox regression *p* = 0.011 Cohort 3—U: Log-rank *p* < 0.0001; M: ns

Abbreviations: AUC = area under the curve; av. = average, CAPRA-S = Cancer of the Prostate Risk Assessment Score; CRPC = Castration-Resistant Prostate Cancer; GS = Gleason Score; HR = hazard ratio; IDC/C = Intraductal Carcinoma and Cribriform Architecture; M = multivariate analysis; NA = not applicable; non-sig = nonsignificant; ns = not specified; OR = odds ratio; pAUC = partial area under the cure; PCa = prostate cancer; PMR = percent methylated ratio; PSA = prostate- specific antigen; RT = radiotherapy; U = univariate analysis. Definitions: BCR: Biochemical recurrence: PSA elevations ≥ 0.2n g/mL post-RP, except [130] > 0.4 ng/mL and [131] > 0.07 ng/mL; clinical recurrence = local recurrence or metastatic relapse; good prognosis: organ-confined disease (pT2) and lack of BCR for at least 5 years; local recurrence: cancer observed on prostatic bed, confirmed by histological analysis of biopsies; metastatic relapse: metastatic deposits (visceral, bony metastasis) confirmed by positive biopsies or cT/bone scans; metastatic-lethal progression = metastatic relapse or PCa death; pathological T-stage: tumour staging based on pathological examination of surgically removed prostate tissue; PCa death: prostate cancer-specific death; poor prognosis: systemic presence of metastatic disease, indicated by recurrence within 3 years and no response to local radiation therapy; progression: either of BCR, metastatic relapse or PCa death; recurrence: either of BCR, local recurrence or metastatic relapse. ^a^ All studies are on prostate cancer tissues from radical prostatectomy, unless specified. ^b^ MethylLight is a quantitative methylation-specific PCR (qMSP) platform. ^c^ Univariate (U) or Multivariate (M) analyses. ^d^ Plus (+) sign indicates variables in the same multivariate model or methylation score together. ^e^ Square bracket ([]) indicate the clinicopathological factors adjusted for in each multivariate model ^f^ The use of bracketed numbers; e.g., (1), (2), indicates different genes, sets of genes or multivariate models validated in the respective study. ^g^ Number in brackets following HR or OR indicate the 95% confidence interval. ^h^ Bolded headings within the table subdivide the different types of genome-wide platforms (restriction-based, capture-based or microarray-based) used in these studies.

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
