# Peer review of "Advances in Prognostic Methylation Biomarkers for Prostate Cancer"

_cancers, 2020, doi:10.3390/cancers12102993_

Round 1

Reviewer 1 Report

The manuscript of Lam et colleagues entitled “Advances in prognostic methylation biomarkers for prostate cancer” gathers many studies dealing with the possibility to use DNA methylation as new biomarkers for prostate cancer in order to guide treatment and monitoring. This work is major and very well written. Few minor points can be discussed:

-Many reviews in the literature have proposed a summary of epigenetic regulations in prostate cancer. In order to differentiate from these reviews, may the authors propose specific guidelines for clinicians to treat or to monitoring aggressive/indolent prostate cancer according to DNA methylation levels/results and involved genes?

-I suggest to the authors to add a list of abbreviations at the beginning of the manuscript.

Author Response

Reviewer #1:

Comments and Suggestions for Authors

The manuscript of Lam et colleagues entitled “Advances in prognostic methylation biomarkers for prostate cancer” gathers many studies dealing with the possibility to use DNA methylation as new biomarkers for prostate cancer in order to guide treatment and monitoring. This work is major and very well written. Few minor points can be discussed:

-Many reviews in the literature have proposed a summary of epigenetic regulations in prostate cancer. In order to differentiate from these reviews, may the authors propose specific guidelines for clinicians to treat or to monitoring aggressive/indolent prostate cancer according to DNA methylation levels/results and involved genes?

Response: Our review is different from other reviews on epigenetic biomarkers in prostate cancer as it focuses specifically on prognostic biomarkers to differentiate indolent and aggressive prostate cancer, rather than diagnostic biomarkers to detect the presence of the disease. Much of the current prognostic biomarker findings are preliminary, or at the pre-clinical stage, and so more work is needed to validate these biomarkers before guidelines can be developed for clinical use. This includes standardization of laboratory methods (technical reproducibility), and the determination of sensitivity and specificity cut-offs.

We have now added a sentence to ‘Future Directions’ to address the reviewer’s comment on ‘specific guidelines’: lines 685-688: ‘Ultimately, the development of specific guidelines for clinical use still requires extensive validation of the best candidate genes in a range of tissue types in independent cohorts with long-term follow-up, for determination of methylation level cut-offs and prognostic validation.’

-I suggest to the authors to add a list of abbreviations at the beginning of the manuscript.

Response: In response to this suggestion, we have added a list of abbreviations. We have included this at the end of the manuscript (Line 697) so as not to interrupt the beginning of the manuscript and disrupt the line numbering; however, this could be inserted where the editors feel is most fitting.  

Reviewer 2 Report

The authors present a comprehensive review of studies investigating methylation biomarkers for prostate cancer prognosis.  For the most part, the manuscript is well written, providing a clear assessment of the field to date.  I have a few fairly minor comments/suggestions.

  1. In the Introduction (line 58), the authors say that a percentage of men on active surveillance develop a "biochemical recurrence". This is not actually the case as the disease isn't recurring (as it would after surgery for instance), it is progressing from a low risk cancer to one that is of higher risk and requires treatment. Also, surgery isn't the only option at this point and some patients will undergo radiation therapy.
  2. In section 2.1, the authors discuss the criteria they used to identify candidate genes, but then in line 167 they present the number of studies that meet these criteria. I would suggest the authors present the number of genes instead and restructure/rewrite the paragraph accordingly.  Or if they prefer to present the number of studies, change the second sentence in this paragraph to reflect this.
  3. Line 217, add "an" to "and as an independent predictor". 
  4. Whilst there is a nice balance in the amount of detail presented in the text vs. the table for most studies, there seems to be a lot of information presented in the text for the study by Litovkin et al. (lines 223-228). Are the authors able to remove some of this information and instead refer the reader to the table?
  5. The terminology CpG "units" seems unusual and rarely used in the literature. Is there a particular reason for using this terminology instead of CpG sites?
  6. Line 242, replace "and" with "or" ?
  7. In the section discussing the PITX2 candidate gene, the authors present the study by Vanaja et al. but don't mention how the methylation score with PITX2 performs.  Can a bit more information be provided?
  8. In the same PITX2 section, the description of the study by Ahmad et al. (ref 98), is very detailed and could be reduced (simply refer the reader to the table).
  9. Line 312, add "factors" to "clinicopathological factors conducted".
  10. Line 318, remove "methylation".
  11. There are a few instances in Table 2 and in the text where only a few of the genes have been mentioned and I'm curious to know why some genes have been left out or why only particular genes have been mentioned? For example, in line 484, why are only ALOX12 and PDGFRB mentioned for the GoldenGate microarray study?  I see that these genes are also candidates in a subsequent study by Angulo et al. but this needs to be mentioned in the text if that is why they are highlighted over other genes. This is also the case for the Mundbjerg study (lines 534-535). In other instances in Table 2, the top 20 genes are presented for the study by Toth et al. but you list 23 genes and for the Geybels study you list 26 of the 32 genes... why not list them all when you list so many? I think there needs to be some consistency across the Table (and text).  Either decide to list those that come up in other studies or just list the first 20 (or a number you prefer) and refer the readers to the original manuscripts for more information.
  12. When discussing the limitations of the field to date in the Conclusions and Future Directions section, there is no mention of the fact that most of the studies have been based on Caucasian or European ancestry samples/populations.  There have been very few studies in other populations and, while some of these markers look promising in EA populations, we don't know how well they will translate to other ethnicities.
  13. Another limitation of the array-based studies that is not mentioned, is the fact that the majority appear to be based on RP samples.  If I've interpreted Table 2 correctly, only one validation study by Savio et al. investigated candidate prognostic markers in diagnostic biopsy samples. Ideally, clinicians would want to apply prognostic biomarkers at diagnosis to help inform treatment strategies, so it is important to determine whether they are also prognostic in such sample cohorts.

Author Response

Reviewer #2:

Comments and Suggestions for Authors

The authors present a comprehensive review of studies investigating methylation biomarkers for prostate cancer prognosis.  For the most part, the manuscript is well written, providing a clear assessment of the field to date.  I have a few fairly minor comments/suggestions.

  1. In the Introduction (line 58), the authors say that a percentage of men on active surveillance develop a "biochemical recurrence". This is not actually the case as the disease isn't recurring (as it would after surgery for instance), it is progressing from a low risk cancer to one that is of higher risk and requires treatment. Also, surgery isn't the only option at this point and some patients will undergo radiation therapy.

Response: Thank you – we have corrected this error: Line 59: “However, 13-45% of low-risk men on active surveillance exhibit a PSA rise and progress to surgery or other treatments [20-22], indicating that they may have been inappropriately assigned to monitoring, and should have been treated earlier.”

  1. In section 2.1, the authors discuss the criteria they used to identify candidate genes, but then in line 167 they present the number of studies that meet these criteria. I would suggest the authors present the number of genes instead and restructure/rewrite the paragraph accordingly.  Or if they prefer to present the number of studies, change the second sentence in this paragraph to reflect this.

Response: We apologise for the confusion, we have now tried to clarify this in the manuscript: Line 169-170: “The studies reporting on these genes are detailed in Table 1.”

  1. Line 217, add "an" to "and as an independent predictor". 

Response: Thank you – we have corrected this error (Line 222).

  1. Whilst there is a nice balance in the amount of detail presented in the text vs. the table for most studies, there seems to be a lot of information presented in the text for the study by Litovkin et al. (lines 223-228). Are the authors able to remove some of this information and instead refer the reader to the table?

Response: We have now removed some of this information accordingly and referred the reader to Table 1: Line 228-230 now reads: “Litovkin and colleagues found trichotomised GSTP1 methylation to be an independent prognostic predictor (when adjusted for GS, pathological T-stage and pre-op PSA levels) of clinical failure (see Table 1 for definition) in two cohorts (Training: n = 147, Validation: n = 71) of high-risk PCa patients [94].”

  1. The terminology CpG "units" seems unusual and rarely used in the literature. Is there a particular reason for using this terminology instead of CpG sites?

Response: In MassARRAY mass spectrometry (Sequenom) technology, the DNA is cleaved into fragments, and the defined analytic units are referred to as ‘CpG units’, which contain either one individual CpG site or an aggregate of consecutive CpG sites. Thus CpG units is the correct terminology to use for methylation data generated using Sequenom analysis (Ehrich M, et al. Proc Natl Acad Sci U S A. 2005. PMID: 16243968).

  1. Line 242, replace "and" with "or" ?

Response: In Line 245, we have replaced “and” with “or”. (We note the line numbers in the current version of the manuscript are slightly different to the reviewer’s line numbering as quoted here).

  1. In the section discussing the PITX2 candidate gene, the authors present the study by Vanaja et al. but don't mention how the methylation score with PITX2 performs.  Can a bit more information be provided?

Response: More information has now been provided on how well the methylation score, which includes PITX2 methylation, performs: Line 301-305 now reads: “Vanaja and colleagues constructed a methylation score consisting of 11 CpG units across 5 genes (from the EpiTYPER MassARRAY platform, see Table 1) including sites in the PITX2 promoter region to predict recurrence within 5 years, in a model combined with GS, pre-op PSA, seminal vesicle involvement and margin status, achieving an AUC of 0.852 (Sensitivity/Specificity = 80/81.2%) [81].”

  1. In the same PITX2 section, the description of the study by Ahmad et al. (ref 98), is very detailed and could be reduced (simply refer the reader to the table).

Response: The description of the Ahmad et al. in the PITX2 section has now been reduced: Line 305-309 now reads: “In a more recent study investigating PCa death as the clinical endpoint in a large cohort of patient-derived TURP tissue (n = 385), a prognostic model was built on six methylation biomarkers (see Table 1) including PITX2, able to improve on the sensitivity of the Cancer of the Prostate Risk Assessment (CAPRA) score to predict aggressive PCa at 10 years follow-up with an AUC of 0.74 (Table 1) [98].”

  1. Line 312, add "factors" to "clinicopathological factors conducted".

Response: “factors” has now been added to Line 314

  1. Line 318, remove "methylation".

Response: “methylation” has now been removed from Line 321

  1. There are a few instances in Table 2 and in the text where only a few of the genes have been mentioned and I'm curious to know why some genes have been left out or why only particular genes have been mentioned? For example, in line 484, why are only ALOX12 and PDGFRB mentioned for the GoldenGate microarray study?  I see that these genes are also candidates in a subsequent study by Angulo et al. but this needs to be mentioned in the text if that is why they are highlighted over other genes. This is also the case for the Mundbjerg study (lines 534-535). In other instances in Table 2, the top 20 genes are presented for the study by Toth et al. but you list 23 genes and for the Geybels study you list 26 of the 32 genes... why not list them all when you list so many? I think there needs to be some consistency across the Table (and text).  Either decide to list those that come up in other studies or just list the first 20 (or a number you prefer) and refer the readers to the original manuscripts for more information.

Response: We apologise for this confusion – in the cases where only a few particular genes have been highlighted, this is because the genes were chosen for further validation or were highlighted as example DMRs with prognostic value in the original text. For example, the GoldenGate study selected ALOX12 and PDGFRB for further functional validation, and we have now clarified this in Table 2: “"PHYMA" signature: 55 probes targeting CpG loci of 46 genes, including at ALOX12, PDGFRB which were selected for functional validation”. Another example is the Bhasin et al. 2015 Cell Rep study, which highlighted five examples of their 101 DMRs in their manuscript (CD14, PCDHGA11, EYA1, CCDC8, HOXC4). In the case of the Mundjberg study, although 25 probes were used in the aggressiveness classifier, we have only listed 18 genes in Table 2 as two probes are within the same gene, and 6 other probes were in intergenic regions.

We agree with the reviewer that there should be consistency across the Table, so for studies reporting a large number of genes we have now only listed the first 20 genes in Table 2 (Angulo et al, Geybels et al, and Toth et al.), and have referred readers to the original studies for further information (Line 360-363): Table 2 highlights the novel biomarkers that were further validated within the original or subsequent studies, and details the top 20 genes for studies that found a large number of significantly associated markers. We refer readers to the original studies for the full lists of methylation markers.”

  1. When discussing the limitations of the field to date in the Conclusions and Future Directions section, there is no mention of the fact that most of the studies have been based on Caucasian or European ancestry samples/populations.  There have been very few studies in other populations and, while some of these markers look promising in EA populations, we don't know how well they will translate to other ethnicities.

Response: Thank you for pointing this out, certainly most studies have been based on Caucasian populations and this is indeed a limitation – we have now added in a sentence into the Conclusions and Future Directions to address this: Line 665-669: “Another limitation in this field is the predominant focus on using Caucasian or European ancestry based populations, with only a handful of studies to date investigating non-Caucasian patients [96,176]. More ethnically diverse populations need to be investigated for discovery of population specific prognostic markers, as well as for examining how well promising biomarkers found in Caucasian populations translate across other ethnicities.”

  1. Another limitation of the array-based studies that is not mentioned, is the fact that the majority appear to be based on RP samples.  If I've interpreted Table 2 correctly, only one validation study by Savio et al. investigated candidate prognostic markers in diagnostic biopsy samples. Ideally, clinicians would want to apply prognostic biomarkers at diagnosis to help inform treatment strategies, so it is important to determine whether they are also prognostic in such sample cohorts.

Response: The reviewer is correct – we have added in a new sentence into the Conclusions and Future Directions to emphasise this point: Line 684-688: “Key to the successful implementation of prognostic biomarkers is the ability to apply them in diagnostic samples, such as needle biopsy or liquid biopsy samples. Ultimately, the development of specific guidelines for clinical use still requires extensive validation of the best candidate genes in a range of tissue types in independent cohorts with long-term follow-up, for determination of methylation level cut-offs and prognostic validation.”

Reviewer 3 Report

Dear authors, 

this is an extensive and excellent review about biomarkers in prostate cancer. The paper was a lot of work. Congratulations.

As a clinical urologist I m most interested in practical aspects of how tests in your summarized studies were performed and might become part of our clinical routine. Especially we need to know which specimen / tissue exactly were used for which clinical question (biopsy tissue, tissue from radical prostatectomy, lymph nodes, selected tumor tissue, urine or even blood)

I think this is of high importance especially as we know that prostate cancer is at least histopathological a very heterogeneous disease and results might depend very much on the specimen.

If possible I therefore suggest that you focus more on that in your summaries and for example ad information about that in your tables. 

Author Response

Reviewer #3:

Comments and Suggestions for Authors

Dear authors, 

this is an extensive and excellent review about biomarkers in prostate cancer. The paper was a lot of work. Congratulations.

As a clinical urologist I am most interested in practical aspects of how tests in your summarized studies were performed and might become part of our clinical routine. Especially we need to know which specimen / tissue exactly were used for which clinical question (biopsy tissue, tissue from radical prostatectomy, lymph nodes, selected tumor tissue, urine or even blood)

I think this is of high importance especially as we know that prostate cancer is at least histopathological a very heterogeneous disease and results might depend very much on the specimen.

If possible I therefore suggest that you focus more on that in your summaries and for example ad information about that in your tables. 

Response: We agree with the reviewer that the type of specimen used in prostate cancer studies is important - we have already noted in the footnote of Table 1 (Line 197) and Table 2 (Line 390) that the studies in these tables were all radical prostatectomy-based studies unless otherwise specified. This is also outlined in the main text: Section 2, Line 135-137: “The studies reviewed in this section have been primarily performed on RP tissue, and studies that have used other types of prostate specimens will be noted accordingly.”  We have now added the following two sentences to ensure this is clear to readers: Line 170-172: “The majority of these studies were performed on RP-derived tissue, when other types of tissue were used (for example, needle biopsies of prostate tissue or urine) this is specified below or within Table 1.”; Line 358-360: “In this section, we summarise genome-wide prognostic methylation biomarker discovery studies in primary PCa tumours, with all but one study performed on RP tissue (Table 2).” Furthermore, we also have a dedicated section to studies investigating methylation biomarkers in non-invasive liquid biopsy specimens, specifically in blood and urine samples (Section 3: Lines 572-639).